# ANALYZING THE LANGUAGE OF VISUAL TOKENS

## ABSTRACT

With the introduction of transformer-based models for vision and language tasks, such as LLaVA and Chameleon, there has been renewed interest in the discrete tokenized representation of images. These models often treat image patches as discrete tokens, analogous to words in natural language, learning joint alignments between visual and human languages. However, little is known about the statistical behavior of these visual languages—whether they follow similar frequency distributions, grammatical structures, or topologies as natural languages. In this paper, we take a natural-language-centric approach to analyzing discrete visual languages and uncover striking similarities and fundamental differences. We demonstrate that, although visual languages adhere to Zipfian distributions, higher token innovation drives greater entropy and lower compression, with tokens predominantly representing object parts, indicating intermediate granularity. We also show that visual languages lack cohesive grammatical structures, leading to higher perplexity and weaker hierarchical organization compared to natural languages. Finally, we demonstrate that, while vision models align more closely with natural languages than other models, this alignment remains significantly weaker than the cohesion found within natural languages. Through these experiments, we demonstrate how understanding the statistical properties of discrete visual languages can inform the design of more effective computer vision models.

## 1 INTRODUCTION

Transformer-based models have not just advanced, but fundamentally reshaped how we approach both vision and language processing, merging these domains in shared sequential representation spaces. Indeed, most recent multi-modal models including DALL-E (Ramesh et al., 2022), LLaVA (Liu et al., 2024) and Chameleon (Team, 2024) operate over joint tokenized representations of images and language, where models decompose images into "visual languages": linearized discrete patches or tokens analogous to words in a sentence. This process, shown in Figure 1, enables seamless integration of images into transformer architectures and allows models to solve multimodal tasks, ranging from image generation and image captioning to visual question answering and translation.

Despite the success of such shared-structure models, current research lacks an in-depth understanding of whether the internal structure of visual tokens mirrors the principles governing natural languages. Specifically, the question arises: do languages formed of visual tokens follow the same statistical patterns, such as frequency distributions, grammatical rules, or semantic dependencies, that human languages exhibit? Investigating such statistical behavior of discrete visual tokens extends beyond theoretical curiosity; it has broad implications for practical machine learning applications. While in linguistic theory, phenomena like Zipf's law and entropy shape natural languages' structure and shape the design of machine learning algorithms, no such rules exist for visual languages. Such rules, if they exist, have the potential to motivate modality-specific models and procedures to capture the unique statistical properties of the underlying visual data.

In pursuit of such rules, in this paper we inspect the equivalence of visual and natural languages through an empirical analysis of token distributions, segmentation granularity, and syntactic and semantic structures. We start by investigating the frequency statistics of visual words and compare them to natural languages. Our analysis reveals that although visual languages can follow power-law (Zipfian) distributions, they use more tokens more uniformly. This leads to languages with greater per-token entropy and lower compression ratios, and implies that vision models may require more attention heads, larger embeddings, and longer training times with more diverse data compared to natural language models (subsection 2.2, subsection 2.3, subsection 2.4, subsection 2.5). Noting in these experiments that visual languages have coarser granularity than patches, we demonstrate through correlation analysis that visual tokens operate at an intermediate level of granularity, and typically represent object parts rather than whole objects or sub-parts in images.

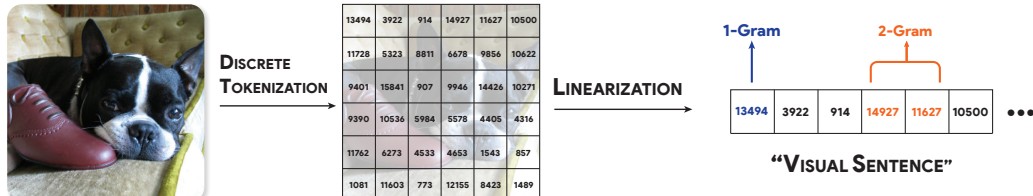

Figure 1: Discrete tokenizers used for visual pre-processing induce "visual languages" made up of sentences containing 1-D sequences of discrete tokens extracted from the images in a dataset. In this paper, we explore how the statistics of these "visual languages" differ from "natural languages," and understand the implications of such statistical differences.

Correspondingly, we show visual tokens are less effective at representing fine-grained details or whole-object structures (subsection 2.6). Following this line of reasoning, we explore if tokens have composable structure, and using parse trees generated by Compound Probabilistic Context-Free Grammars (C-PCFG), we show visual languages have grammatical structures that are more fragmented, with grammars trained on them exhibiting higher perplexity compared to natural languages (section 3). We then confirm these observations by building a co-occurrence based embedding space, and evaluating the topological alignment between natural and visual languages. In this, we find visual languages align more with natural languages than with other visual languages, but less so than natural languages align with each other (subsection 3.1).

Together, we aim to show through these experiments that while visual languages have striking similarities to natural languages, there are also notable and fundamental differences, motivating unique modality-specific approaches to vision-language learning.

## 2 DO VISUAL TOKENS ACT LIKE WORDS?

The first question that we examine is: Do visual tokens themselves (i.e. the patches of an image) act like words? While we often treat these tokens as either a word (or subword), as each token forms a single input sequence element in a transformer, it seems unintuitive that there would be a one to one statistical correlation between the two concepts. In this section, we look at several statistical properties of individual tokens, comparing those observed in natural language to those in visual systems.

### 2.1 PRELIMINARIES

What, explicitly, is a visual language? In this work, we consider a visual language to be a language induced over "visual tokens" by first converting images in a dataset to a discrete set of symbols using a visual tokenizer (often a VQ-VAE), and then linearizing those tokens into one-dimensional sequences (See Figure 1). Such a definition parallels efforts in both text-to-image diffusion and large vision and language models which have both explored using discrete visual tokens for vision-language model alignment (Team, 2024; Ramesh et al., 2022; Gu et al., 2022; Razavi et al., 2019), as well as in uni-modal models such as LVM (Bai et al., 2024) and LLamaGen (Sun et al., 2024).

We primarily focus on common tokenizers used for recent vision and language models, and our selection of tokenizers is overviewed in Table 1. These tokenizers are all VQ-VAE-based, trained on varying datasets, and with various methods. While some recent models such as Transfusion (Zhou et al., 2024) and LLaVA Liu et al. (2024) leverage continuous-valued tokens instead of discrete vocabularies, there is still considerable uncertainty about whether discrete or continuous-valued tokens are more effective (Mao et al., 2021). While many of our methods in this paper could apply to continuous tokens through a discrete quantization of those tokens, we leave such continuous extensions to future work. For more details on the tokenizers, see Appendix A.

We ground our empirical experiments in several common multi-modal datasets, including Conceptual Captions (12M) (Sharma et al., 2018), MS-COCO (Lin et al., 2014), ILSVRC (ImageNet) (Russakovsky et al., 2015) and XM-3600 (Thapliyal et al., 2022). Each of these datasets has a set of images, and (except ILSVRC) paired text in one or more languages. For more information on the datasets, see Appendix B. An example visual sentence from MS-COCO (Image ID: 399655) is given in Figure 1. In all of the experiments in this paper, we linearize the tokens using a row-wise scan order (for a detailed discussion on scan-order, see Appendix C). Such linearization is the de facto standard for turning spatial visual tokens into sequences of discrete tokens for use in learning applications.

| Tokenizer | Application | Resolution | Vocab Size |
|---|---|---|---|
| chameleon-512 (Team, 2024) | Multimodal Foundation Model | $512 \times 512$ | 8192 |
| compvis-vq-f8-64 (Rombach et al., 2022) | Image Generation | $64 \times 64$ | 16384 |
| compvis-vq-f8-256 (Rombach et al., 2022) | Image Generation | $256 \times 256$ | 16384 |
| compvis-vq-imagenet-f16-1024-256 (Esser et al., 2021) | Image Generation | $256 \times 256$ | 1024 |
| llamagen-vq-ds16-c2i (Sun et al., 2024) | Text $\rightarrow$ Image | $256 \times 256$ | 16384 |

Table 1: Visual tokenizers that we use in this paper. We select several tokenizers across several applications at varying resolutions and vocab sizes.

## 2.2 TOKEN FREQUENCY AND ZIPF'S LAW

The statistics of natural language token distributions have long been studied, beginning with Dewey (1921), who first plotted the frequency of English words. A key principle that emerged from this research is Zipf's Law (Kingsley Zipf, 1932), which describes a power-law relationship between the frequency of words and their rank in a language where a small number of high-frequency words dominate natural language, while the majority of words occur infrequently. Formally, Zipf's law states that:

$$f(r) \propto r^{\alpha + \sigma Z} \tag{1}$$

where $f(r)$ is the frequency of the element with rank $r$ and $\alpha/\sigma$ parameterize a learned Gaussian distribution (close to 1/0 in many natural languages).

Zipf's law has been observed across many languages (Gelbukh & Sidorov, 2001; Yu et al., 2018) and non-human communication systems (such as dolphins (McCowan et al., 1999)). As Mandelbrot pointed out, adherence to Zipf-like distributions ensures that communication systems—whether natural or artificial—operate efficiently (Mandelbrot, 1953). Language models, especially large language models (LLMs), have been shown to follow this same pattern, with token distributions that obey Zipf's law (Patwary et al., 2019). This statistical regularity in language extends beyond word frequency - Zipf's law has also been observed in images themselves: Ruderman (1997) showed that the distribution of object sizes and spatial frequencies in natural scenes follows power-law distributions, and Crosier & Griffin (2007) showed that there was Zipfian behavior in image coding schemes such as JPEG.

Thus, we first ask the question - **Do "visual languages" follow Zipf's law?** To do this, we tokenize the image datasets according to subsection 2.1 and compute the empirical token-rank frequency distributions on each of the datasets (See Appendix D for details). We show the empirical distributions in Figure 2. If the plots were Zipfian, we would expect them to be linear in the log-log space; while this is the case for natural languages, visual languages do not seem to generally conform to a linear curve, instead, for one and two grams, the plots follow a lognormal distribution, and for higher level N-grams are more convex in nature.

For one/two-grams, this indicates that token utilization is fairly uniform, with most tokens occurring in equal proportion, and the heavier tails of the distribution indicate that "rare" are, in practice, not so rare, occurring with much higher frequency than expected under a power-law distribution. Whereas natural languages are often structured with a clear core vocabulary and then more specialized words, it seems like visual features seem to be more evenly distributed, with many features or combinations being equally likely. At higher n-grams, for visual languages there is more convex behavior, suggesting that there are very few common n-grams, instead, n-grams are often unique, and composed in ways that appear very infrequently within the datasets. Such an implication implies that visual languages are highly context-dependent (which is sensible, as visual scenes are quite complex).

To confirm these details, we fit a Zipf's distribution to each of the models, with the results of the fit shown in Table 2. Interestingly, the $\alpha$ values have opposite behaviors for visual and natural languages in the light of increasing $N$. In natural languages, the fact that $\alpha$ increases with $N$ means that higher-order N-grams follow steeper power-law distributions, and the distribution of N-gram frequencies becomes more concentrated around a few common combinations, while the frequency of rare combinations decreases rapidly. In visual languages, on the other hand, the decrease in $\alpha$ with increasing $N$ suggests that higher-order combinations of visual features follow flatter distributions: as visual N-grams increase in complexity, there is more diversity in the combinations of features and patterns, leading to richer and more distributed sets of higher-order feature combinations.

These phenomena together suggest that VQ-VAEs are "spreading" information between the independent tokens, rather than building compressive and compositional structures, which we explore further in subsection 2.2 (token innovation) and subsection 2.5 (compression). Indeed, since Zipf's Law reflects a (theoretically optimal) balance between redundancy and information, it suggests that visual languages are more data-driven, and reflect the underlying complexity and variability of visual scenes, rather than focusing

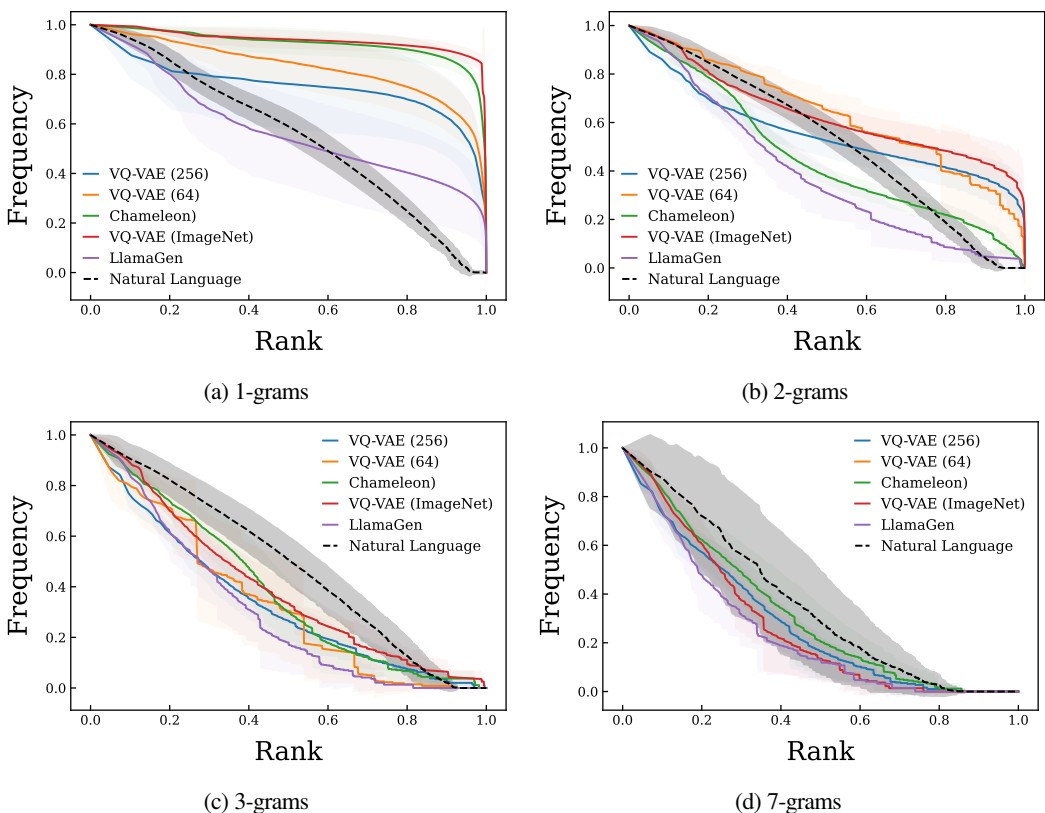

Figure 2: Plot of normalized token log-frequency against normalized Log-Rank for several visual and textual languages for different n-grams, aggregated across datasets. While the tails of visual languages do not conform to Zipf's law well for small values of $N$, for larger values of $N$, the fit becomes more linear.

|  | N=1 | | N=2 | | N=3 | | N=5 | | N=7 | |
|---|---|---|---|---|---|---|---|---|---|---|
|  | Natural | Visual | Natural | Visual | Natural | Visual | Natural | Visual | Natural | Visual |
| $\alpha$ | $1.71_{0.23}$ | $4.37_{1.33}$ | $1.99_{0.25}$ | $4.43_{1.50}$ | $2.28_{0.33}$ | $2.57_{0.82}$ | $2.85_{0.82}$ | $2.35_{0.52}$ | $3.02_{0.73}$ | $2.35_{0.50}$ |
| $\sigma$ | $0.01_{0.02}$ | $0.18_{0.14}$ | $0.01_{0.01}$ | $0.07_{0.16}$ | $0.03_{0.04}$ | $0.09_{0.18}$ | $0.25_{0.54}$ | $0.09_{0.13}$ | $0.28_{0.46}$ | $0.09_{0.14}$ |
| $\overline{\log\mathcal{L}}$ | $-4.03_{1.38}$ | $-9.72_{3.28}$ | $-3.11_{0.41}$ | $-4.24_{2.27}$ | $-2.92_{0.42}$ | $-3.53_{1.68}$ | $-2.43_{0.67}$ | $-2.99_{1.22}$ | $-1.98_{1.02}$ | $-2.72_{1.19}$ |

Table 2: Comparison of aggregate power law fit metrics ($\alpha, \sigma$, mean log-likelihood) across different N-gram lengths for natural and visual languages. While visual languages do not follow Zipf's law for $N=1$, the fit is significantly better for $N=3$ and above.

on reducing redundancy for communicative operations. Such a deviation might suggest that models that are more Zipfian, such as chameleon, may be better placed as embedding/alignment models for visual tasks, whereas models such have more convex N-gram distributions are better for high-fidelity generation tasks.

Beyond model quality/applicability implications, the fact that visual languages don't follow Zipf's Law implies that traditional NLP-inspired techniques (e.g., those relying on power-law distributions such as compression algorithms, or memory-based systems based on Zipfian patterns) may not directly apply to visual languages. Beyond this, visual languages likely require different optimization techniques taking into account the non-linear distribution of N-grams – methods that handle long-tail distributions might be more appropriate than techniques focused on heavy tails. Such differences in distribution could also suggest that higher-order interactions between visual features are more important in vision models than in language models, and model architectures should be designed to capture these higher-order patterns effectively.

### 2.3 TOKEN INNOVATION

One thing that stands out from the experiments in subsection 2.2 is that single visual tokens appear more uniformly than single words, inspiring the question: do new images consist of mostly new tokens, or

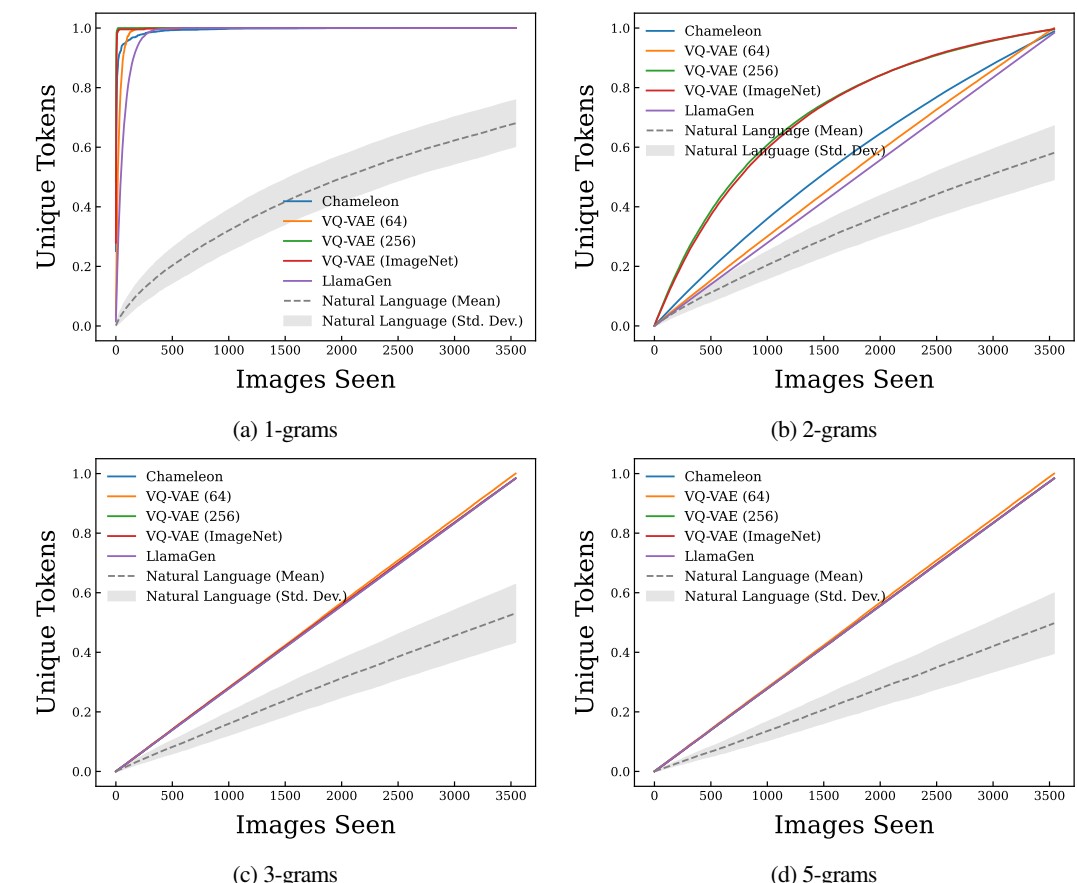

(a) 1-grams

(b) 2-grams

(c) 3-grams

(d) 5-grams

Figure 3: Comparison of unique tokens as a function of images seen on the XM-3600 dataset for different N-grams. While higher values of N approach a linear relationship in the visual languages, textual languages are always sub-linear in their growth. Surprisingly, for 3/5-grams, several visual language curves overlap.

do new images re-combine existing tokens in novel ways? In natural language, this has generally been codified by Heaps'/Herdan's law (Herdan, 1964; Heaps, 1978), which says that vocabularies' sizes are concave increasing power laws of texts' sizes (See Appendix E for details).

To explore this effect, Figure 3 plots the number of unique tokens seen against the number of images seen for the XM-3600 dataset for several visual tokenizers and natural languages. The natural languages follow the expected distribution, with unique tokens increasing sub-linearly with respect to the number of images. The visual tokens, on the other hand, appear much more rapidly. For single tokens, almost all of the tokens in the vocabulary appear within the first 100 images, suggesting that the rate of token innovation is significantly higher than that of natural languages. For 2-grams and 4-grams, the relationship trends linear, but never approaches the sub-linear behavior that is expected of generative systems which follow Heaps' law. Additional experiments on MS-COCO are given in Appendix E.

We further fit a Yule-Simon distribution (Simon, 1955) to both the natural and visual languages. The Yule-Simon process is a stochastic model for generating sequences of words or tokens, where the probability of introducing a new token decreases as more tokens are added, leading to a power-law distribution; mathematically, this process is governed by a probability proportional to the current token frequency, combined with a parameter that controls the rate of new token introduction (see Appendix F for more details). The results, given in Figure 4 and Appendix F, demonstrate that the generative process for new tokens largely does not fit with that described by the Yule-Simon process in the visual case, however, fit quite well for many text languages.

The fact that visual tokens have a much higher rate of innovation has several key implications for the design, training, and evaluation of both generative and discriminative models. The high vocabulary diversity of visual tokens means that while generative models will be able to generate higher-fidelity output, discriminative models are at high risk of over-fitting: risking overly specific captions or inconsistencies across similar

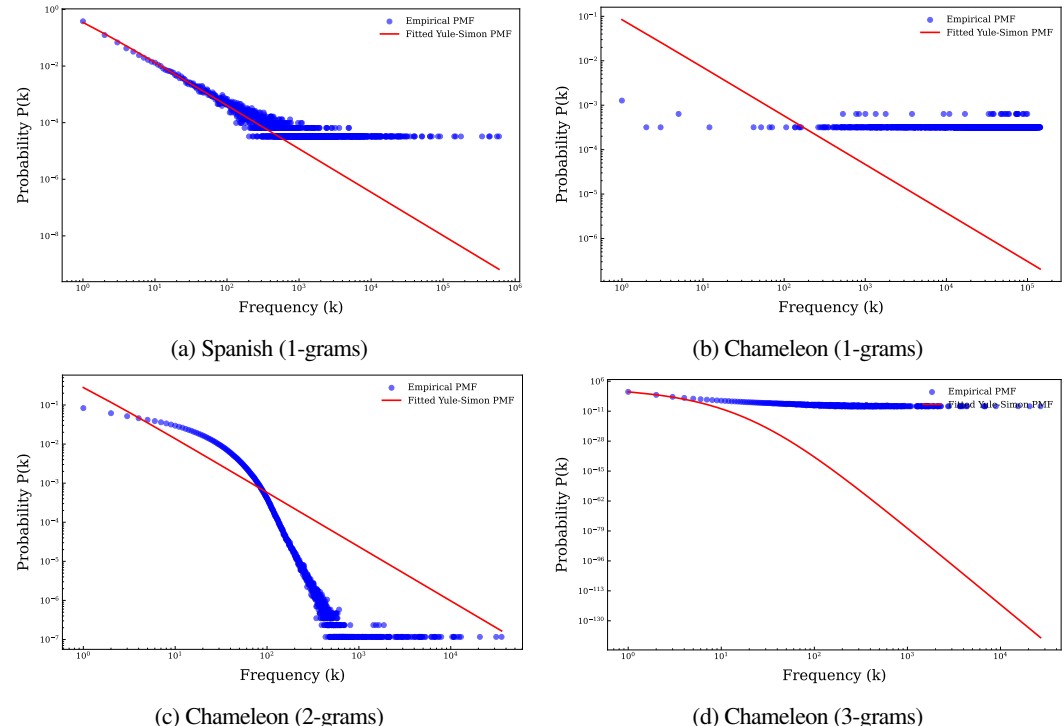

(a) Spanish (1-grams)  (b) Chameleon (1-grams)

(c) Chameleon (2-grams)  (d) Chameleon (3-grams)

Figure 4: Yule-Simon model fit for Chameleon vs. Spanish on the COCO dataset (More models/languages in Appendix F). While Spanish (and in general, natural languages) largely fits a Yule-Simon model, Chameleon does not appear to be generated by such a process at any n-gram level.

images (a feature that has already been noted in several works (Chan et al., 2022; Caglayan et al., 2020)). Such high vocab diversity also impacts the training efficiency of models: both generative and discriminative models will require longer training times and need more varied datasets to handle expanding token sets than models of natural language (a fact which has been observed explicitly in (Touvron et al., 2021), and more generally with vision transformers). Beyond training, inference and evaluation are also impacted. Decoding approaches that rely on frequency/presence penalties may want to leverage unique/more aggressive penalties for vision compared to language tokens. For evaluation, perhaps already clear from existing work, semantic-based evaluation is likely more effective than token-based evaluation in visual approaches due to the high level of diversity in the local token space (Anderson et al., 2016; Hessel et al., 2021).

## 2.4 NATURALITY

Benford's Law describes the frequency distribution of leading digits in naturally occurring datasets, where smaller digits like 1 and 2 appear disproportionately more often than larger digits like 8 and 9 (Benford, 1938). Originally observed in domains such as physics (Sambridge et al., 2010), economics (Tödter, 2009), and demographics (Miller, 2015), recently, there has been growing interest in extending this statistical principle to linguistic data (Golbeck, 2023; Melián et al., 2017; Hong, 2010). One of the primary applications of Benford's law is the detection of anomalies in data: datasets that do not follow Benford's law are likely to be unnatural in nature - here, we ask the question, do visual language token frequencies naturally follow Benford's law? We follow a similar tokenization process to subsection 2.2, and plot the occurrence of leading digits in the token frequency distribution (See Appendix G for more details).

Our results are shown in Figure 5 for the MS-COCO dataset, and in Appendix G on other datasets. Interestingly, for single tokens, the distribution is unique-token-heavy, with the remaining tokens having a Gaussian distribution around six. Two-grams are the most natural, with Chameleon following Benford's law almost exactly, with three-grams significantly dominated by low/unique frequency tokens. Interestingly, the highest quality tokenizer, the Chameleon tokenizer, is by far the most natural in Figure 5a, suggesting that tokenizations performing well for vision-text tasks might have more natural distributions. Beyond this effect, Figure 5b shows that distributions of visual-token bi-grams have the most natural distribution curves, implying a potential correspondence in statistics between visual bi-grams and text uni-grams,

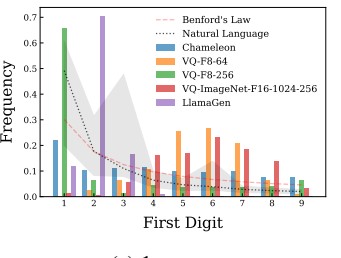 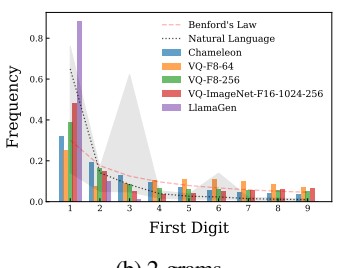 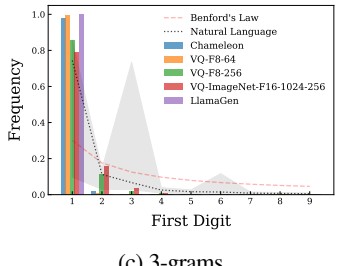

|  (a) 1-grams | (b) 2-grams | (c) 3-grams |

Figure 5: Plot of the first digits of the token frequency distribution on the MS-COCO dataset. While 1-grams have a uniquely 1-heavy head, with a Gaussian tail (around 6), 2-grams naturally follow an exponential decay function, and 3-grams are dominated by unique tokens. The grey area represents the maximum and minimum among the 36 natural languages.

Table 3: Understanding the entropy and Huffman compression rates of visual and natural languages ($p < 0.01$ across all metrics). While the compression rate improves slightly with two-grams in the visual case, it is reduced significantly in the natural case. Full results in Table H.1.

| Language | Avg Code Length | Entropy | Fixed Code Length | Orig Bits (M) | Huff Bits (M) | Comp. Rate | % Reduction |
|---|---|---|---|---|---|---|---|
| Visual | $10.7 \pm 1.9$ | $10.7 \pm 1.9$ | $11.0 \pm 1.8$ | $5.35 \pm 1.3$ | $5.20 \pm 1.3$ | $1.03 \pm 0.02$ | $2.9 \pm 1.9$ |
| Visual (N=2) | $18.1 \pm 0.8$ | $18.1 \pm 0.8$ | $18.7 \pm 0.5$ | $9.1 \pm 1.5$ | $8.8 \pm 1.5$ | $1.03 \pm 0.02$ | $3.2 \pm 2.2$ |
| Natural | $9.0 \pm 0.9$ | $8.9 \pm 0.9$ | $13.8 \pm 0.9$ | $4.10 \pm 3.1$ | $2.54 \pm 1.8$ | $1.55 \pm 0.1$ | $34.9 \pm 6.1$ |
| Natural (N=2) | $13.5 \pm 1.0$ | $13.5 \pm 1.0$ | $16.3 \pm 1.1$ | $4.9 \pm 3.8$ | $3.9 \pm 3.0$ | $1.21 \pm 0.08$ | $16.9 \pm 5.2$ |

and suggesting that future work in tokenization could explore vocabularies of token bi-grams or bi-gram compression for vision tokenizers.

## 2.5 ENTROPY AND REDUNDANCY

Building on the foundational work of Shannon (1951), entropy and redundancy have long been understood as key characteristics of natural language, providing insight into its inherent predictability and compressibility. While natural languages, like English, exhibit high redundancy that enables efficient encoding, it is unclear if visual languages might have similar coding behaviors. To evaluate the efficiency of encoding, we use a similar setup to subsection 2.2 and extract token streams for each of the target datasets. We then compute the entropy of the token streams, as well as compute a simple Huffman code/compression (Huffman, 1952) for each of the resulting streams. Such a hierarchical compression code allows us to estimate the overall "compressibility" of the stream (See Appendix H for background/details).

The results are summarized in Table 3. We can see that in general, the average code length, entropy, and bits of information/sample are higher for visual languages. This suggests that visual languages have more variability and are inherently more complex to predict and encode than natural language. This is unsurprising, given the complexity and richness of the visual world, compared to the sparsity of natural language, however, it is somewhat surprising that the entropy is not massively different from natural languages, suggesting that visual tokenizers are capable of reducing the richness of natural language to suitably sparse representations for reasoning. Notably different is the "compressibility" of the token streams. While natural language tokens are highly compressible using Huffman encoding, visual languages are almost incompressible, suggesting that information is highly distributed amongst the tokens and that there is very little structural reuse between the different images. While we explore grammars further in section 3, this experiment indicates that it is unlikely that models have non-trivial grammars of tokens, instead, these tokens are more local, and particularly high-variance.

These experiments have several potential implications for model design. First, since visual tokens have significantly higher entropy and lower compressibility, it may be necessary to use more attention heads, deeper models, and more dense embeddings, in visual-based models in order to capture a sufficient number of relationships and higher-level representations of visual information. Models like LLaVA (Liu et al., 2024) with simple projection layers between the visual token and text token spaces may not perform as well on downstream visual tasks as models such as mPlug (Ye et al., 2024) which have more dense transformer-based adapters (a result which is empirically verified by Tong et al. (2024), who leverage a spatially aware dense connector to achieve significant performance improvements).

Table 4: Whole, part, and sub-part purity/part-normalized mutual information on the SPIN dataset. PP: Part Purity (%), VTP: Visual-Token Purity (%), PNMI: Part-Normalized Mutual Information.

| Tokenizer | Wholes | | | Parts | | | Sub-Parts | | |
|---|---|---|---|---|---|---|---|---|---|
| | PP | VTP | PNMI | PP | VTP | PNMI | PP | VTP | PNMI |
| chameleon-512 | 2.512 | 0.216 | 1.557 | 4.399 | 0.138 | 0.256 | 1.660 | 0.200 | 0.898 |
| compvis-vq-f8-64 | 3.061 | 0.526 | 6.148 | 5.653 | 0.308 | 1.760 | 2.611 | 0.508 | 6.246 |
| compvis-vq-f8-256 | 2.333 | 0.467 | 0.925 | 4.209 | 0.334 | 0.122 | 1.527 | 0.426 | 0.434 |
| compvis-vq-imagenet | 2.467 | 0.739 | 1.463 | 4.354 | 0.479 | 0.207 | 1.626 | 0.623 | 0.787 |
| llamagen-vq-ds16-c2i | 4.384 | 0.107 | 13.711 | 6.983 | 0.057 | 4.487 | 3.656 | 0.112 | 13.273 |

It's worth noting that Huffman encoding is independent of the scan order of the images, and instead, focuses only on token frequencies. It would be interesting for future work to explore how scan order impacts compress-ability, and we discuss potential experiments and limitations regarding scan order in Appendix C.

### 2.6 TOKEN SEGMENTATION GRANULARITY

One common question for many vision researchers is: "do visual tokens represent objects?" Indeed, while visual tokens are spatially fixed to patches in the image, because of the VQ-VAE training process, it is un-clear if they take on additional non-spatial semantic meaning. Recently, Hsu et al. (2021) demonstrated that in audio domains, HuBERT tokens (audio-tokens) have relatively high mutual information with phoneme representations of audio, suggesting that self-supervised models are capable of learning natural structures despite being segmented to fixed-width patches. Can we answer this question for visual languages as well?

Recently, Myers-Dean et al. (2024) introduced the SPIN dataset, a new labeled dataset of hierarchically segmented objects, where the objects are labeled at the whole, part, and sub-part levels. This gives us per-image annotations of the existence of wholes, parts, and sub-parts. From this, we compute several measures of natural correlation between these part-annotations and the visual token languages, inspired by Hsu et al. (2021) (For more details, see Appendix I): **Part Purity**, a metric that measures the average accuracy of assigning a visual-token to its most likely part label, reflecting image-level part consistency within a particular visual token, **Visual-Token Purity**, a metric that assesses how well images containing the same part label are consistently assigned to the same visual-tokens and **Part-Normalized Mutual Information**, an information-theoretic metric which measures the percentage of uncertainty about the part-label eliminated after observing a particular visual token.

The results are summarized in Table 4. In general, tokenizers appear to be most effective at capturing part-level representations, as evidenced by consistently higher Part Purity (PP) values for parts compared to wholes or sub-parts across all models. This suggests that tokenizers are better aligned with mid-level structures (parts), rather than whole objects or fine-grained sub-parts. However, Visual-Token Purity (VTP) remains low across all models and levels of granularity, indicating that images containing the same part-label are not consistently assigned to the same visual tokens, reflecting fragmentation in the clustering. PNMI values are generally higher for sub-parts than for parts or wholes, particularly in models like `llamagen-vq-ds16-c2i`, which shows the highest PNMI across all levels. This implies that tokenizers can capture more fine-grained information at the sub-part level, though the corresponding decrease in part purity for sub-parts suggests that while they can reduce uncertainty about part labels, their actual clustering of sub-parts is inconsistent.

## 3 ARE VISUAL LANGUAGES STRUCTURED LIKE NATURAL LANGUAGES?

In subsection 2.5 we showed that visual languages are not very compressible using Huffman encodings, suggesting that visual languages may not have hierarchical structures similar to those of natural languages. To inquire further into this question, we test whether Context-free Grammars (Chomsky & Schützenberger, 1959) can approximate the structure of visual languages as well as they can natural languages by fitting grammars to each modality using unsupervised grammar induction techniques.

Particularly, we use Compound Probabilistic Context-Free Grammars (C-PCFG) (Kim et al., 2019) as the grammar formalism for our experiments. C-PCFGs are a type of neural PCFG, where grammar production rules are modeled as compound probability distributions (Robbins, 1956) – every production depends on both the set of symbols in the grammar as well as a global latent variable $z$. This formulation, trained with variational methods, allows for global sentence information to flow through all parsing decisions in a sentence while remaining compatible with efficient inference methods which standard PCFGs enjoy (Baker, 1979). For more details on C-PCFGs see Appendix J.2.

| Dataset | PPL | PPL-R | MBF | FR | CU |
|---------|-----|-------|-----|-----|-----|
| COCO-DE | 24.70 | 99.61% | 3.00 | 2.44 | 1.00 |
| COCO-EN | 25.18 | 99.62% | 3.02 | 2.43 | 1.00 |
| COCO-VQ | 671.93 | 95.80% | 1.41 | 1.75 | 0.97 |
| XM3600 | 739.37 | 95.40% | 6.65 | 6.39 | 0.33 |
| CC12M | 595.01 | 96.28% | 2.85 | 2.54 | 1.00 |
| ILSVRC | 654.25 | 95.92% | 1.93 | 2.28 | 0.93 |
| SPIN | 656.61 | 95.89% | 1.27 | 1.82 | 0.73 |

(a) Generated parse tree statistics

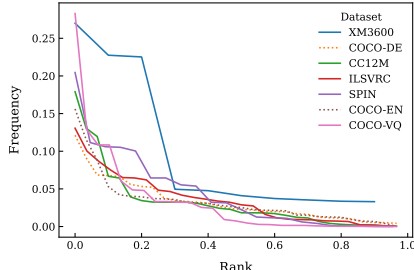

(b) Parse tree non-terminal node frequencies

Figure 6: Comparison between C-PCFG grammars trained on textual and visual languages. Grammars learned over text exhibit greater reduction in perplexities (PPL-R) with comparable parse tree heights (FR), right-branching propensity (MBF), Non-terminal codebook utilization (CU), and non-terminal node label frequencies (b).

C-PCFG memory costs are cubic on sentence length, leading us to use the `compvis-vq-f8-64` tokenizer for visual grammars, which provides a tractable 32 tokens per image. For each dataset, we train grammars over five seeds for 15 epochs and select the seed with the lowest test set perplexity for analysis. We test our pipeline by evaluating parsers learned on English COCO captions (COCO-EN) against silver-label parse trees extracted with Benepar (Kitaev & Klein, 2018), attaining an F1 score of 49 on the best seed, which is comparable to prior work (Zhao & Titov, 2020).

We report test set statistics over learned grammars, such as final parse tree perplexity (PPL) and percentage reduction in perplexity (PPL-R) from random initialization to convergence. The mean branching factor (Li et al., 2024) (MBF) measures on average whether generated parse trees tend to branch right or left. This is achieved by averaging the proportion of leaves between the right and left branches of nodes $n$ across parse trees $t$ in the dataset:

$$\text{MBF}(t) = \frac{1}{|t|} \sum_{n \in t} \frac{\text{CR}(n)}{\text{CL}(n)} \qquad (2)$$

Here, CR and CL represent the counts of leaves in the right and left branches of a node, respectively. To get a better sense of parse tree topology, we also measure the ratio between tree height (the length of the longest path in the tree) and the minimum possible height for the tree:

$$\text{FR}(t) = \frac{\text{H}(t)}{\log \text{L}(t)} \qquad (3)$$

Where H($t$) and L($t$) are the height and number of tokens in the input sequence, respectively. Codebook utilization (CU) measures the percentage of non-terminal labels utilized within generated parse trees.

We present these statistics in Figure 6a, as well as normalized non-terminal node frequencies for parse trees generated by each grammar in Figure 6b, with some example parse trees in Figure J.1. Although both modalities experience a great reduction in perplexity compared to random initialization, textual grammars (COCO-EN and COCO-DE) generally exhibit greater reductions in perplexities than visual grammars, corroborating findings from subsection 2.5 which suggest that visual tokens are not as compressible as textual tokens. Although visual grammars converge to PPL values an order of magnitude greater than the textual grammars, we observed that their PPL values at the start of training are proportionally higher, likely due to the generally longer visual sentence length (32 tokens in these experiments). All other measures are generally comparable across modalities – both modalities show similar proclivities towards right-branching trees (MBF), although visual grammars are somewhat more balanced. Both modalities present similar tree heights (FR), with the non-terminal label codebooks being largely utilized. The notable exception to these trends is the grammar trained on XM3600 tokens. XM3600 contains a significantly lower number of training examples (one order of magnitude less than SPIN, and two orders less than all other datasets), which may have resulted in a degenerate grammar being learned.

These results suggest that the structure of visual languages may not be as well approximated by context-free grammars as natural languages are. This raises the question of whether they may be better fit by other grammatical formalisms, such as mildly context-sensitive grammars (Yang et al., 2023) which allow for dependencies to cross between token spans.

Table 5: Summary of Procrustes/Hausdorff alignment distances between vision languages and natural languages on the MS-COCO dataset. While in general, all languages are poorly co-aligned, in general, vision languages align slightly, but significantly, more strongly with natural languages than they do with other vision models.

| Distance | Language | Language-to-Vision Distance | Language-to-Natural Language Distance | Closest Language | Closest Distance |
|---|---|---|---|---|---|
| **Procrustes** | Natural (Average) | $0.96689 \pm 0.00425$ | $0.96530 \pm 0.00735$ | text-hr | 0.96333 |
| | Chameleon | $0.97699 \pm 0.0158$ | $0.96474 \pm 0.00382$ | text-no | 0.95580 |
| | VQ-VAE (256) | $0.97886 \pm 0.01427$ | $0.96532 \pm 0.00401$ | text-ko | 0.95381 |
| | VQ-VAE (64) | $0.97875 \pm 0.01517$ | $0.97024 \pm 0.00310$ | text-it | 0.96329 |
| | VQ-VAE (ImageNet) | $0.97896 \pm 0.01478$ | $0.96731 \pm 0.00386$ | text-hu | 0.95709 |
| **Haussdorf** | Natural (Average) | $10.81174 \pm 1.28073$ | $9.42697 \pm 1.15902$ | text-pl | 7.60177 |
| | Chameleon | $7.68661 \pm 0.58783$ | $6.56173 \pm 0.38642$ | text-ko | 5.85738 |
| | VQ-VAE (256) | $7.24126 \pm 0.27003$ | $5.95511 \pm 0.34980$ | text-zh | 5.36121 |
| | VQ-VAE (64) | $7.97373 \pm 0.29399$ | $6.90376 \pm 0.42660$ | text-it | 6.02011 |
| | VQ-VAE (ImageNet) | $7.52335 \pm 0.24214$ | $5.91748 \pm 0.58758$ | text-hr | 4.92164 |

## 3.1 TOPOLOGICAL SIMILARITY

To expand our discussion on structural similarity, we further investigate how similar the topological structures of visual and textual tokens are, and whether these similarities can reveal meaningful insights about the underlying representations, i.e. can we observe strong structural alignment points between the natural and visual latent spaces, or are there notable deviations?

We begin by training GloVe embeddings (Pennington et al., 2014) on co-occurrence matrices derived from visual tokens and textual tokens present in the captions (details in Appendix J). This gives us a continuous topology of similar dimension within which we can explore potential alignment. We then explore two pairwise distance matrices between the two GloVe vector spaces: Procrustes alignment (Gower, 1975) and directed Haussdorf distance (Bowen, 1979).

Figure J.2 gives the Procrustes similarity and Figure J.3 gives the directed Haussdorf distance between the models, with some key aggregates summarized in Table 5. While there are few clear trends, a key finding is that vision models are largely more aligned with natural language models than they are with each other, with Chameleon being slightly more central than other models (perhaps due to its training process). Overall, the lack of strong alignment trends between different vision models highlights that their latent spaces are more fragmented, suggesting that visual token representations are often model-specific or task-dependent, rather than universally structured. Notably, however, some languages align much better with visual models than others (such as Korean to the Chameleon tokenizers, or Hungarian/Polish in general), suggesting that some tokenizers may be significantly stronger when aligning to specific languages. Another interesting observation is that the directed Hausdorff distance shows that the natural language to vision model alignment is significantly further than the vision model to natural language alignment. This results implies that generation of images from text is much harder than the generation of text from images - something often observed in practice.

Given the overall distances between these structural representations, our experiments suggest that future model architectures should focus on reducing this asymmetry. Specialized models that effectively encode multimodal information - and perhaps aligned tokenization methods (such as CLIP), represent promising future directions for research.

## 4 CONCLUSION

This paper takes a first look at visual languages from the angle of empirical statistics. While there are similarities between how we currently treat visual and natural languages/sentences - the experiments in this paper show that, at least statistically, visual tokens and natural languages are far from trivially aligned. Such poor statistical alignments motivate both unique model architectures and training procedures for visual transformers (summarized tabularly in Appendix K) - and we hope that this work inspires further research into novel architectures, designs, and hyper-parameters for vision-token based models. Indeed, while some of the hypotheses that we outlined in this paper have already been demonstrated, many of the suggestions (such as increasing frequency penalties when decoding visual languages) remain untested in practice - and it is interesting and necessary future work to close the loop on such potential modifications. We hope, as a whole, that this work inspires additional research into fundamental statistics as a motivation for new architectural decisions and directions.

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

APPENDIX

The appendix consists of the following further discussion:

- Appendix A discusses the tokenizers used when constructing the visual languages, with detailed descriptions of Chameleon, Stable Diffusion, and LlamaGen tokenizers.
- Appendix B describes the datasets utilized in this work, including Conceptual Captions (CC12M), MS-COCO, ImageNet (ILSVRC), XM-3600, and SPIN.
- Appendix C describes potential limitations and opportunities for future work.
- Appendix D describes the Zipf experiments in subsection 2.2, and gives additional experimental details.
- Appendix E describes Heaps' law, and gives additional experimental results to complement subsection 2.3.
- Appendix F explains the Yule-Simon distribution, the methodology used to fit this distribution to observed token frequencies, and the experimental results from token frequency analysis.
- Appendix G discusses the process used for analyzing visual tokens according to Benford's law in subsection 2.2, including n-gram extraction and first-digit distribution analysis across datasets.
- Appendix H explains the Huffman encoding experiments, measuring entropy and compression efficiency of tokenized visual data.
- Appendix I explores segmentation granularity and how visual tokens correspond to parts and sub-parts in images, using co-occurrence metrics like Part Purity and Visual Token Purity.
- Appendix J discusses CPFCGs, the process for extracting GloVe embeddings from both vision and language tokenizers, and the topological analysis used in subsection 3.1.
- Appendix K clearly enumerates the implications of our work from a model-design and training perspective.

## A  TOKENIZERS

In this work, we explore three families of VQ-VAE (Van Den Oord et al., 2017) based tokenizers for images. While the general details are given in Table 1, we expand on the details for the tokenizers here.

**Chameleon (Team, 2024):** Chameleon is a family of early-fusion token-based mixed-modal models capable of understanding and generating images and text. The image tokenizer, `chameleon-512`, is based on Gafni et al. (2022), which is a modified VQGAN Esser et al. (2021) model which adds perceptual losses to specific image regions such as faces and salient objects (in an attempt to improve the fidelty of generated images). The chameleon tokenizer is trained from scratch on a closed-source set of licensed images, and encodes images at a resolution of $512 \times 512$ into a discrete token codebook size of 8192 and dimension 256. Notably, when training the tokenizer the model up-samples the percentage of images with faces by two times to improve performance on human face generation (which may somewhat skew the performance of the tokenizer on non-face based images).

**Stable Diffusion (Compvis) (Rombach et al., 2022):** Stable Diffusion is a latent text-to-image diffusion model, which learns a joint distribution over image and text representations in a discretized latent space. Similar to the chameleon tokenizer, these tokenizers are trained in an adversarial manner following Esser et al. (2021) on OpenImages Kuznetsova et al. (2020), such that a patch-based discriminator can differentiate original images from reconstructions. The stable diffusion tokenizers (`compvis-vq-f8-64` and `compvis-vq-f8-256`) have an image resolution of $384 \times 384$ with a crop-size of 256, and use a codebook dimension of size 4, with a very high VQ quantization dimension of 16384. While these models were trained at a crop size of 256, for grammatical analysis, many of the generated sequences are much too long to solve using traditional methods. Thus, we additionally consider a model, `compvis-vq-f8-64` which uses a $64 \times 64$ crop of the image, which produces linearized sequences of a more manageable length of 32, used in section 3. The tokenizer `compvis-vq-imagenet-f16-1024-256` (originally trained by Esser et al. (2021)) uses the same training procedure as those in Rombach et al. (2022), but was trained on the ImageNet dataset, with a codebook of dimension 256, and size 1024.

**LlamaGen (Sun et al., 2024):** LlamaGen is a family of image-generation models that apply next-token prediction to perform iamge synthesis. The LlamaGen tokenizer, `llamagen-vq-ds16-c2i` takes images

of resolution $256 \times 256$, and uses a codebook of size 16384 and dimension 8. `llamagen-vq-ds16-c2i` is trained on the ImageNet training dataset.

| Tokenizer | R-FID | R-IS | PSNR | PSIM | SSIM |
|---|---|---|---|---|---|
| `chameleon-512` | - | - | - | - | - |
| `compvis-vq-f8-64` | - | - | - | - | - |
| `compvis-vq-f8-256` | 1.14 | 201.92 | 23.07 | 1.17 | 0.650 |
| `compvis-vq-imagenet-f16-1024-256` | 4.98 | - | - | - | - |
| `llamagen-vq-ds16-c2i` | 2.19 | - | 20.79 | - | 0.675 |

Table A.1: Tokenizer Performance (As available in the original papers) - Evaluated on ImageNet 50K Validation dataset.

The three tokenizers examined in this work—**Chameleon**, **Stable Diffusion**, and **LlamaGen**—each employ distinct methodologies and design choices tailored to their respective goals in image representation and synthesis. **Chameleon** is a mixed-modal model designed to improve image fidelity, particularly for faces and salient objects, by up-sampling face images during training and applying perceptual losses to critical regions. It encodes images at a high resolution of $512 \times 512$ into a large codebook of size 8192 and dimension 256, focusing on generating high-quality human face representations (which may bias the overall results).

**Stable Diffusion** tokenizers, by contrast, emphasize flexible image synthesis through adversarial training on diverse datasets such as OpenImages and ImageNet. Their design includes smaller image resolutions ($384 \times 384$ or cropped to $64 \times 64$ or $256 \times 256$) and an exceptionally large VQ quantization dimension of 16384 for robust latent space discretization. This flexibility allows adaptation to various tasks, such as generating more manageable sequence lengths for grammatical analysis. Finally, **LlamaGen**, using next-token prediction for synthesis, applies a more compact structure with a codebook of size 16384 and dimension 8, trained on the ImageNet dataset at a lower resolution ($256 \times 256$). While less focused on high-fidelity synthesis than Chameleon, LlamaGen aims to balance efficiency and performance.

## A.1 Defining N-Grams For Vision Tokenizers

To define N-grams, we follow the procedure indicated in Figure 1 of the paper: tokens are first linearized using a row-wise linearization scheme (as is done in traditional transformer approaches), giving a 1-D sequence of tokens $(x_1, x_2, ..., x_n)$. N-grams are then defined analogously to natural language, with 2-grams being a sequence of all pairs of tokens (i.e. $(x_1, x_2)$, $(x_2, x_3)$, $(x_3, x_4)$, etc.), 3-grams being a sequence of all triplets of tokens (i.e. $(x_1, x_2, x_3)$, $(x_2, x_3, x_4)$, etc.) and other N-grams being defined similarly.

## B Datasets

In this work, we explore the effects of tokenization across several datasets:

**Conceptual Captions (12M) (Sharma et al., 2018):** Conceptual captions (12M, CC12M) is a dataset with approximately 12 million image-text pairs soruce from web alt-text, traditionally used for vision-language pre-training.

**MS-COCO (Lin et al., 2014):** The MS-COCO dataset is a dataset for image description containing 328K images, each with 5 ground truth descriptions in English. In addition to the standard annotations, we also leverage translated annotations from Thapliyal et al. (2022), which provide machine translations into 36 languages for each of the MS-COCO images.

**ImageNet (ILSVRC) (Deng et al., 2009):** ImageNet contains approximately 1.2M images which are manually annotated to indicate the objects present in each image. These annotations are linked to the WordNet hierarchy, providing a rich set of object categories. The dataset covers 1,000 object classes for the classification task, including common objects like animals, vehicles, and household items.

**XM-3600 (Thapliyal et al., 2022):** The Crossmodal-3600/XM3600 dataset is a multilingual multimodal evaluation dataset designed to support image captioning tasks across 36 languages. It consists of 3600 geographically diverse images, each annotated with human-generated captions that are consistent across languages but not derived from direct translations, ensuring linguistic naturalness and cultural relevance. The images were selected from regions where these languages are spoken, drawn from the Open Images Dataset using a careful algorithm to ensure regional diversity.

**SPIN (Myers-Dean et al., 2024):** The SPIN (SubPartImageNet) dataset is a hierarchical semantic segmentation dataset designed to provide detailed annotations for natural images at multiple levels of granularity, specifically focusing on objects, parts, and subparts. SPIN builds on the PartImageNet dataset, expanding

its scope by introducing over 106,000 subpart annotations across 203 subpart categories, covering 34 part categories from diverse objects such as animals, vehicles, and human figures. The dataset contains 10,387 images divided across 11 supercategories, including rigid objects like cars and non-rigid entities like animals.

The datasets explored in this work—**Conceptual Captions (12M)**, **MS-COCO**, **ImageNet (ILSVRC)**, **XM-3600**, and **SPIN**—present unique challenges and opportunities for token-based analysis in vision-language models. Their diversity in scale, annotations, and contextual richness impacts the statistical properties of the visual languages induced by tokenization. These properties are helpful for understanding the analyses in this work, which directly influence the performance of models on multimodal tasks.

For example, the large scale of **Conceptual Captions (12M)** helps provide insights into how diverse image-text pairs impact token entropy and the uniformity of token utilization. **MS-COCO**, alternatively, supports detailed studies of token alignment between visual and linguistic modalities, facilitating evaluations of grammatical induction and cross-modal representations. Additionally, the multilingual nature of the captions provides a testing ground for understanding cultural and linguistic variations in tokenization. **ImageNet (ILSVRC)**, on the other hand, offers a well-structured dataset for studying token representations in object-centric tasks. **SPIN**'s emphasis on hierarchical segmentation of images into objects, parts, and sub-parts allows for detailed analysis of how tokenization captures different levels of semantic granularity, with implications for clustering and information encoding. Finally, the geographically diverse and culturally grounded captions of **XM-3600** facilitate the study of tokenization's adaptability to varying linguistic and cultural contexts, shedding light on its impact on model generalization.

## C   LIMITATIONS

While this paper does have significant empirical results, we want to recognize the several potential limitations/opportunities for future work:

**Tokenizer Selection:** While the paper does focus on a fairly wide range of common (and modern) visual tokenizers, there is a fairly large potential selection of additional tokenizers that could be compared. Indeed, a key limiting factor is that all of the tokenizers explored in this work are VQ-VAE based. As discussed in subsection 2.1, a detailed analysis of continuous tokenizers (such as auto-encoders which are KL-regularized, CLIP-style encoders, or BERT-style encoders) would provide significant additional information. Directly applying natural language statistics to these continuous embeddings, however, is non-trival, as to understand ideas of "token frequency" or "grammar", such analyses would have to either (a) be extended to the continuous domain, or (b) the tokens themselves would have to be quantized to discrete representations. For example, for entropy, continuous domain generalizations exist (such as differential entropy), however are challenging to quantify in higher dimensional spaces, and it remains unclear if such entropy values are comparable to those in the discrete domain. For Benford's law, no such continuous domain generalization exists, and would have to be derived from first-principles. While it appears that there is some intuition as to the underlying foundational principles behind Benford's law (Becker et al., 2018), simply deriving (and demonstrating) such a continuous generalization would be a significant undertaking. Similar techniques would have to be derived for other methods such Yule-Simon laws or C-PCFGs.

While is possible to perform analyses on quantized spaces of the continuous domain, treating the quantized states as continuous variables, however doing so introduces significant quantization bias that can impact the outcomes. For example when analyzing entropy in quantized spaces, the resolution of quantization directly impacts the calculated entropy. Coarse quantization tends to underestimate the entropy by failing to capture the full variability of the continuous domain, while fine quantization can overfit noise in the data. Similarly, for Yule-Simon distributions, the observed frequencies of quantized states would have the potential to reflect artifacts of binning rather than true reflections of the underlying continuous distribution. Thus, the resulting power-law exponent might be systematically distorted, either attenuated or exaggerated, based on the quantization scheme used.

Overall, we believe that such extensions are highly interesting, but are worthy of detailed analysis and discussion which is outside the scope of this initial work.

**Dataset Coverage:** Another limiting factor of this research is the dataset coverage. While it is impossible to analyze all data, visual information is highly diverse, and domains such as medical imaging, geospatial imaging, or autonomous driving may have entirely different statistics. In general, however, we found that across the datasets that we did use (which represent a fairly general slice of traditional training data), the statistical representations were similar. For example, it is fairly challenging to distinguish any dataset-level

patterns in Figure D.1, which shows a per-dataset breakdown of the empirical token frequency distributions, or Figure F.2 which shows the Yule-Simon fits for emprical token frequencies.

**Scan Order of Images:** One of the notable limitations of this work is that we primarily investigate a linear row-wise scan order of the images. We primarily limit our experiments to this scan order as (1) this is the de-facto scan order used in all existing transformer-based tokenization schemes and (2) we do not want to introduce further confounding analytical axes in this work. Exploring non-row-wise scan orders is, however, an extremely interesting question. In our limited experimentation, we found that a row-wise scan order does not significantly impact the explorations in the paper, as the majority of the analyses are scan-order independent.

**Token Granularity and Semantic Understanding:** Although granularity analysis is insightful, a deeper examination of how well visual tokens capture complex semantic meaning in images (e.g., context, object relationships, or scene understanding) remains future research. We strongly believe that future research should explore how visual tokens represent not just parts of objects but also their roles in broader scenes or tasks requiring semantic understanding (e.g., visual reasoning, narrative generation), however such explorations would require signficant new labeled data, or novel statistical approaches.

**Visual Tokens in Video Data:** In tasks like video understanding or motion tracking, the temporal relationships between visual tokens might reveal additional complexities not captured in static image analysis. Future research could explore how the behavior of visual tokens changes in sequential or temporal data settings and whether current statistical patterns hold when accounting for time.

## D    ZIPF'S LAW

As discussed in subsection 2.2, Zipf's Law (Kingsley Zipf, 1932), describes a power-law relationship between the frequency of words and their rank in a language where a small number of high-frequency words dominate natural language, while the majority of words occur infrequently. Formally, Zipf's law states that:

$$f(r) \propto r^{\alpha + \sigma Z} \tag{D.1}$$

where $f(r)$ is the frequency of the element with rank $r$ and $\alpha/\sigma$ parameterize a learned Gaussian distribution (close to 1/0 in many natural languages).

For each dataset and tokenizer, to compute the power law fit, we leverage the method/code in Alstott et al. (2014). When fitting the power laws, because of computational limits, we limit the number of processed N-grams to 5M, and on CC12M and ILSVRC, unless otherwise noted, we compute the n-grams on only a subset of the full dataset consisting of a randomly sub-sampled 200K image set). Results broken down by N-gram are shown in Figure 2, while results broken down by model/dataset are given in Figure D.1

## E    HEAPS'/HERDAN'S LAW

Heaps' law (also referred to as Herdan's law) is an empirical rule that describes the relationship between the size of a corpus and the number of unique word in the corpus (Heaps, 1978; Herdan, 1964). Specifically, it predicts that as the size of a text grows, the number of unique words increases, but at a decreasing rate.

Mathematically, the law is described by:

$$V(N) = kN^{\beta} \tag{E.1}$$

where V(N) is the number of distinct words (the vocabulary size), N is the total number of words, and $k$ and $\beta$ are parameters, $0 < \beta < 1$. Heaps' law reflects the fact that even as new text is added to a corpus, the frequency of newly introduced words diminishes, meaning a large corpus doesn't proportionally expand its vocabulary.

Plots for unique tokens vs. images seen on XM-3600 are given in Figure 3, with those for MS-COCO given in Figure E.1.

## F    YULE-SIMON DISTRIBUTION

The Yule-Simon distribution (Willis & Yule, 1922) is a model often used to describe processes where new elements (in this case, tokens) are introduced over time with a probability that decreases as the existing set of elements grows. Specifically, for a sequence of tokens, the Yule-Simon distribution describes the probability of the $k$-th token occurring $m$ times as:

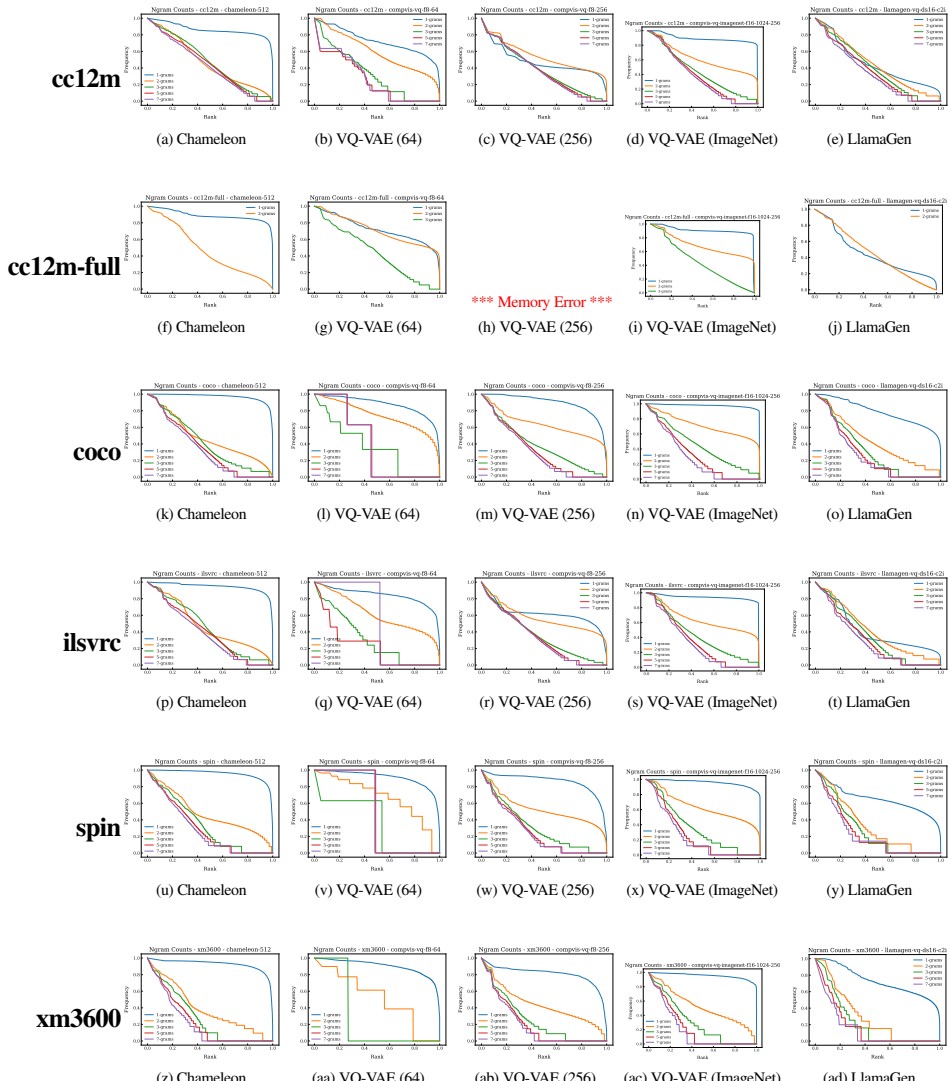

Figure D.1: Empirical N-gram distributions for different datasets comparing normalized log-rank against normalized log-frequency. In general, visual languages do not achieve power-law distributions, and when they do, it is at high levels of N, and fairly steep slopes (compared to natural langauges).

$$P(m) = \alpha \mathrm{B}(m, \alpha + 1) \tag{F.1}$$

where $\mathrm{B}(\cdot, \cdot)$ is the Beta function. This captures the balance between token reuse and token innovation, and the shape parameter $\alpha$ reflects the likelihood of encountering a novel token versus reusing an existing one.

## F.1 EXPERIMENTAL DESIGN

For each dataset and tokenizer configuration on the COCO and XM-3600 datasets, we fit the Yule-Simon distribution to the observed token frequency distributions by minimizing the negative log-likelihood using the L-BFGS-B optimization algorithm. This method is selected due to its ability to handle the bound constraints placed on the parameter $\alpha$, ensuring that $\alpha > 0$. The optimization starts with an initial guess of $\alpha = 1.0$, and the negative log-likelihood is computed based on the observed token frequencies. The optimization process continues until convergence, with the final $\alpha$ value corresponding to the best-fit parameter for the Yule-Simon distribution. Invalid $\alpha$ values are penalized by assigning an infinite log-likelihood to ensure feasible solutions. Once the optimal $\alpha$ is found, we compute the empirical PMF from the frequency distributions by normalizing the observed token counts. In parallel, the theoretical PMF is computed using the fitted $\alpha$ value.

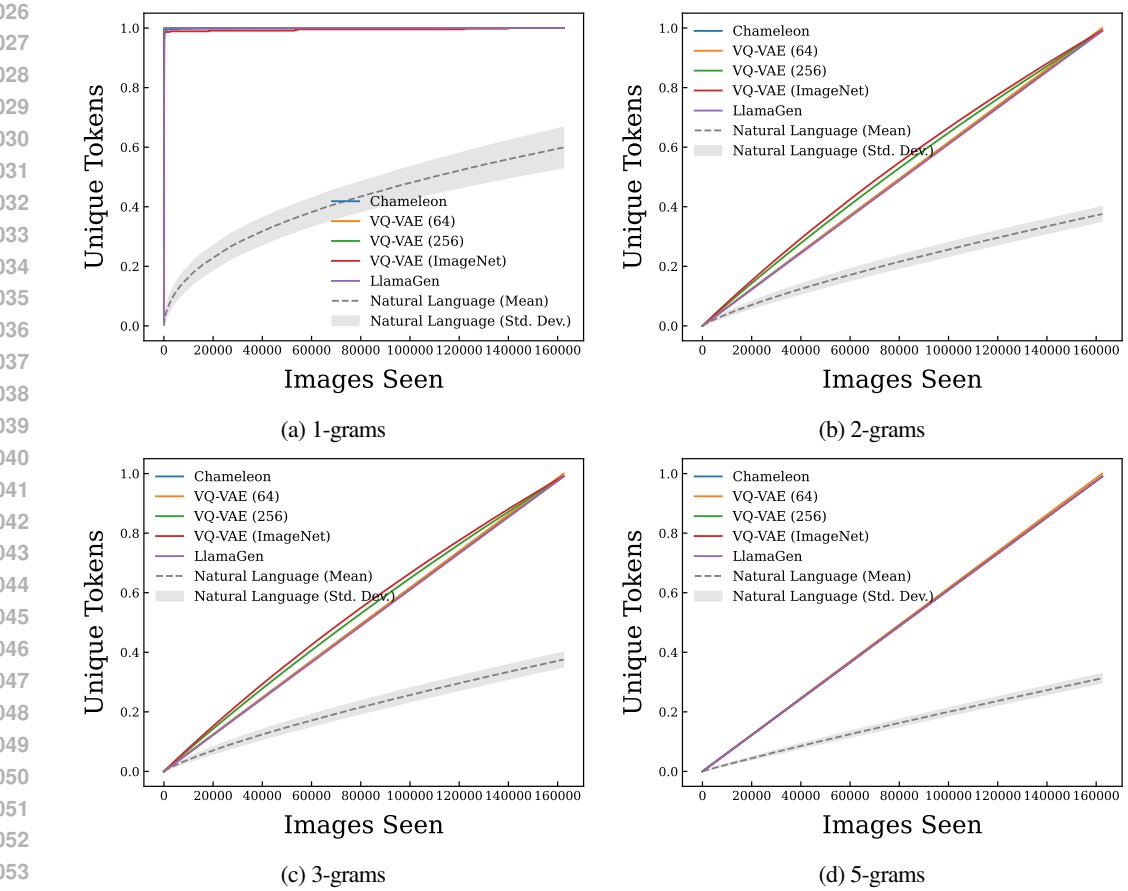

(a) 1-grams

(b) 2-grams

(c) 3-grams

(d) 5-grams

Figure E.1: Comparison of unique tokens as a function of images seen on the MS-COCO dataset for different N-grams.

## F.2 ADDITIONAL EXPERIMENTAL RESULTS

The full experimental results on text data for the XM-3600 dataset are shown in Figure F.1, with model data shown in Figure F.2. The text results for COCO are shown in Figure F.3, with COCO model convergence shown in Figure F.4.

## G BENEFORDS LAW

Benford's Law (Benford, 1938) describes the distribution of leading digits in many naturally occurring datasets, where smaller digits are more likely to appear as the first digit. Specifically, the probability $P(d)$ of a digit $d$ (where $d$ is between 1 and 9) being the leading digit is given by:

$$P(d) = \log_{10}\left(1 + \frac{1}{d}\right) \tag{G.1}$$

According to this law, the number 1 appears as the first digit around 30% of the time, while larger digits like 9 appear less frequently, around 5% of the time.

For each dataset and tokenization configuration, we extract n-grams (with n = 1, 2, and 3) from tokenized text and image data. We aggregate the token frequencies by computing the distribution of the first digits of these counts. Specifically, the first digits of each token frequency are extracted, and their occurrences are counted to form a first-digit distribution. In cases where natural language data is available, we also compute aggregate distributions across multiple locales for text-based tokenizations. The aggregated text distributions include the mean, standard deviation, minimum, and maximum values for each first-digit count across different locales.

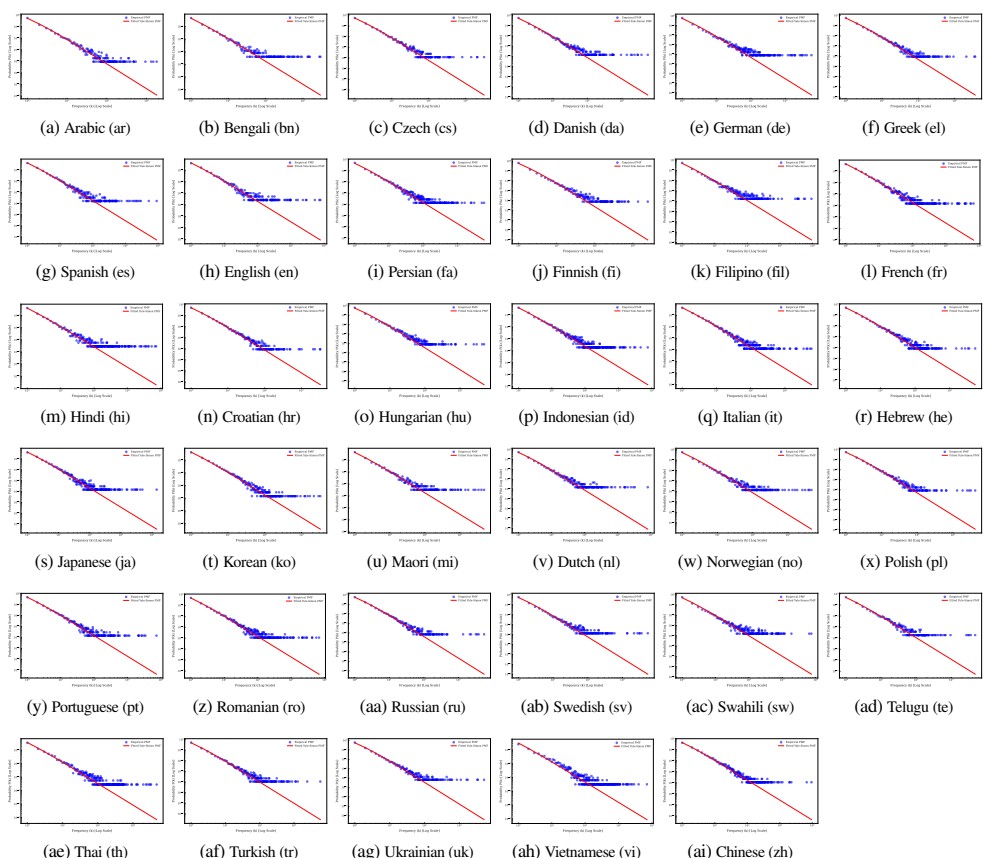

Figure F.1: Log-log fits on XM-3600 for various languages (Simon model, $n{=}1$)

The full results for each of the datasets (XM-3600, CC12M, COCO, ILSVRC and SPIN) is given in Figure G.1.

## H  HUFFMAN ENCODING / ENTROPY

Huffman encoding is a widely-used algorithm for lossless data compression, which assigns variable-length codes to tokens based on their frequencies. The core idea is to minimize the total number of bits required to represent the token stream by assigning shorter codes to more frequent tokens and longer codes to less frequent ones. This is achieved by constructing a binary tree where each token is a leaf, and its depth (or code length) corresponds to its frequency. The encoding process ensures that the total number of bits, $L_{\text{Huffman}}$, needed to encode a stream of tokens is reduced compared to fixed-length encoding, where each token would require $\lceil \log_2(n) \rceil$ bits, with $n$ being the number of unique tokens.

Entropy, denoted as $H(X)$, represents the theoretical limit on the average number of bits needed to encode the token stream, and is calculated using Shannon's entropy formula:

$$H(X) = -\sum_{x \in X} P(x) \log_2 P(x) \tag{H.1}$$

where $P(x)$ is the empirical probability of token $x$ in the stream. In this experiment, entropy serves as a benchmark for comparing the performance of Huffman encoding. The closer the average code length of the Huffman encoding is to the entropy, the more efficient the compression. By evaluating the compression rate and percentage reduction, we can quantify how effectively Huffman encoding reduces the bit length compared to the fixed-length encoding, with the goal of approaching the entropy limit.

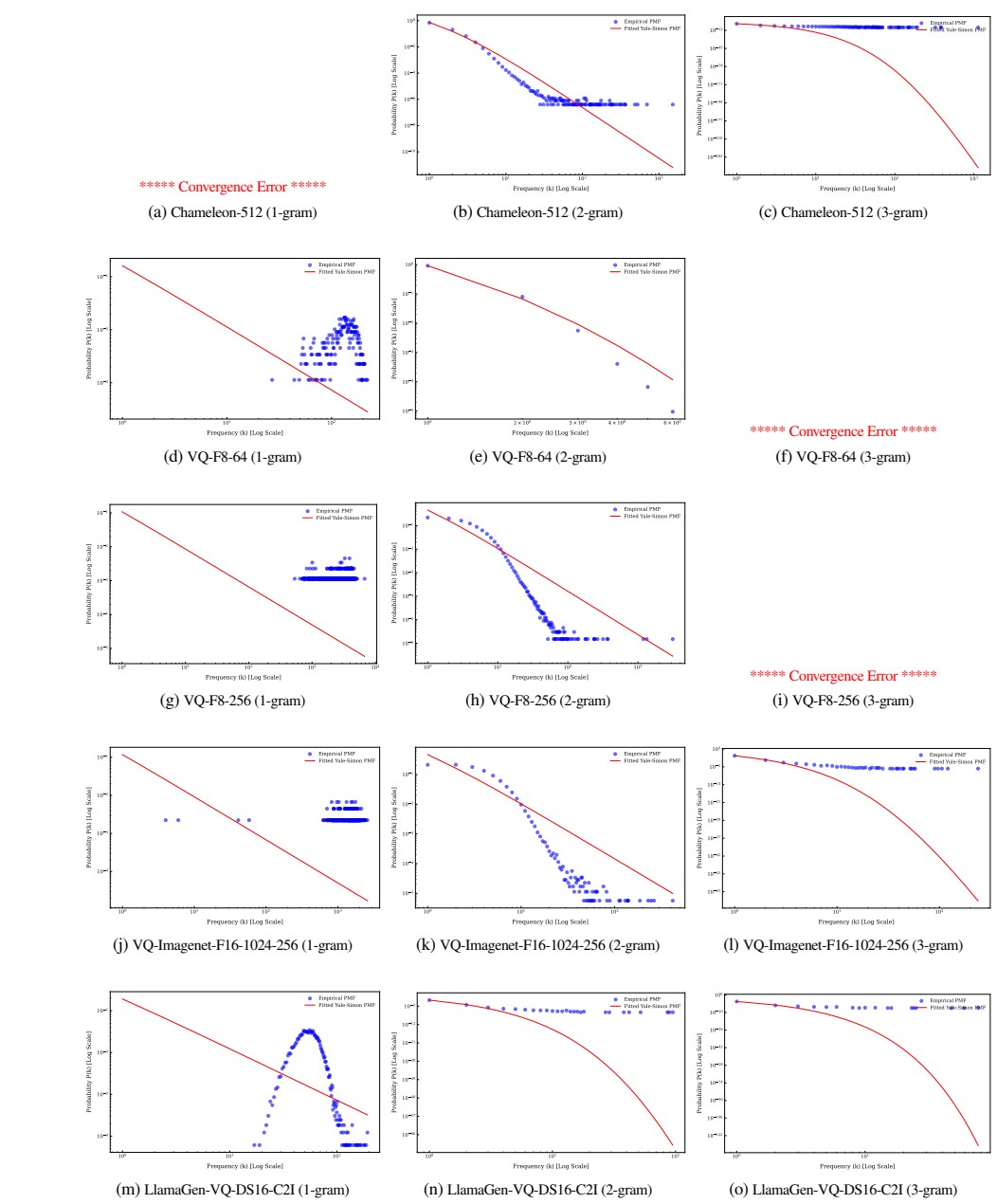

Figure F.2: N-gram analysis on various models (Simon model on XM-3600)

## H.1 EXPERIMENTAL DESIGN

In our experiments, for each dataset/tokenizer combination (both $N=1$ (unigrams) and $N=2$ (bigrams)), we extract the first 500,000 tokens as a token stream, and then apply Huffman encoding to compress this token stream. We extract several key metrics from the resulting encoding:

- Average Code Length: The weighted average of the lengths of Huffman codes for all tokens.
- Entropy: The theoretical minimum average code length for the specified token distribution (See Equation H.1).
- Fixed Code Length: The length of the fixed-length codes used for comparison.
- Original Bits: The number of bits required for fixed-length encoding of the token stream.
- Huffman bits: The number of bits required after applying Huffman encoding.

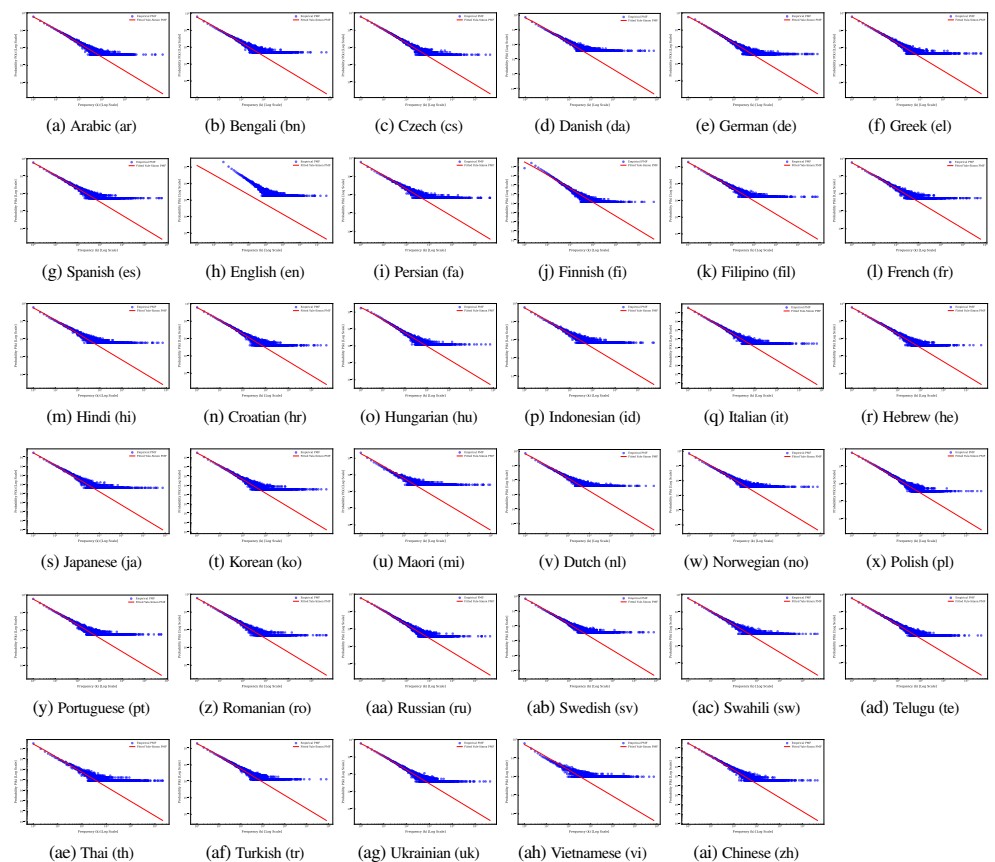

Figure F.3: Log-log fits on COCO for various languages (Simon model, $n = 1$). The horizontal shift in the English frequencies is likely caused by duplicate unfiltered captions in the empirical distribution.

- Compression Rate: The ratio of the original bits to the Huffman bits.
- Percentage Reduction: The percent reduction in the total number of bits after applying Huffman encoding.

## H.2 FURTHER EXPERIMENTAL RESULTS

The full experimental results for the Huffman coding experiment are given in Table H.1. A surprising detailed result is that the chameleon tokenizer, the most effective of the tokenizers, is also the most compressible representation of them, with almost twice the percentage reduction compared to other models. Llama-gen is the least compressible, and indeed, is almost completely incompressible, suggesting it has very efficient token use but does not contain any repeatable structure.

Table H.1: Full huffman coding results for N=1 and N=2. ACL:Average Code Length, E: Entropy, FC: Fixed Code Length, OB: Original Bits (MB), HB: Huffman Bits (MB), CR: Compression Rate, PR: Percentage Reduction

| Dataset | Model | N | ACL | E | FCL | OB | HB | CR | PR |
|---|---|---|---|---|---|---|---|---|---|
| coco | chameleon-512 | 1 | 11.30 | 11.27 | 12 | 6.00 | 5.65 | 1.06 | 5.86 |
| coco | compvis-vq-f8-64 | 1 | 9.78 | 9.74 | 10 | 5.00 | 4.89 | 1.02 | 2.25 |
| coco | compvis-vq-f8-256 | 1 | 9.71 | 9.67 | 10 | 5.00 | 4.85 | 1.03 | 2.93 |
| coco | compvis-vq-imagenet-f16-1024-256 | 1 | 8.80 | 8.76 | 9 | 4.50 | 4.40 | 1.02 | 2.22 |
| coco | llamagen-vq-ds16-c2i | 1 | 13.98 | 13.95 | 14 | 7.00 | 6.99 | 1.00 | 0.18 |
| coco | text-ar | 1 | 9.85 | 9.82 | 15 | 7.50 | 4.93 | 1.52 | 34.31 |
| coco | text-bn | 1 | 8.91 | 8.88 | 14 | 7.00 | 4.46 | 1.57 | 36.36 |
| coco | text-cs | 1 | 9.80 | 9.77 | 15 | 7.50 | 4.90 | 1.53 | 34.64 |
| coco | text-da | 1 | 8.09 | 8.06 | 14 | 7.00 | 4.05 | 1.73 | 42.21 |
| coco | text-de | 1 | 8.70 | 8.67 | 15 | 7.50 | 4.35 | 1.72 | 41.98 |
| coco | text-el | 1 | 8.55 | 8.53 | 14 | 7.00 | 4.28 | 1.64 | 38.90 |
| coco | text-es | 1 | 7.98 | 7.95 | 14 | 7.00 | 3.99 | 1.75 | 42.99 |

| Dataset | Model | N | ACL | E | FCL | OB | HB | CR | PR |
|---|---|---|---|---|---|---|---|---|---|
| coco | text-fa | 1 | 8.20 | 8.17 | 13 | 6.50 | 4.10 | 1.59 | 36.94 |
| coco | text-fi | 1 | 10.14 | 10.11 | 16 | 8.00 | 5.07 | 1.58 | 36.65 |
| coco | text-fil | 1 | 7.47 | 7.44 | 14 | 7.00 | 3.74 | 1.87 | 46.63 |
| coco | text-fr | 1 | 8.12 | 8.09 | 14 | 7.00 | 4.06 | 1.72 | 42.02 |
| coco | text-hi | 1 | 8.22 | 8.19 | 14 | 7.00 | 4.11 | 1.70 | 41.31 |
| coco | text-hr | 1 | 9.66 | 9.63 | 15 | 7.50 | 4.83 | 1.55 | 35.58 |
| coco | text-hu | 1 | 8.89 | 8.87 | 15 | 7.50 | 4.45 | 1.69 | 40.70 |
| coco | text-id | 1 | 8.33 | 8.30 | 13 | 6.50 | 4.17 | 1.56 | 35.91 |
| coco | text-it | 1 | 8.33 | 8.30 | 14 | 7.00 | 4.17 | 1.68 | 40.50 |
| coco | text-he | 1 | 9.90 | 9.87 | 15 | 7.50 | 4.95 | 1.51 | 33.98 |
| coco | text-ja | 1 | 7.87 | 7.83 | 14 | 7.00 | 3.93 | 1.78 | 43.81 |
| coco | text-ko | 1 | 8.70 | 8.67 | 14 | 7.00 | 4.35 | 1.61 | 37.87 |
| coco | text-mi | 1 | 6.83 | 6.80 | 13 | 6.50 | 3.42 | 1.90 | 47.45 |
| coco | text-nl | 1 | 7.96 | 7.93 | 14 | 7.00 | 3.98 | 1.76 | 43.11 |
| coco | text-no | 1 | 8.12 | 8.09 | 15 | 7.50 | 4.06 | 1.85 | 45.87 |
| coco | text-pl | 1 | 9.84 | 9.80 | 15 | 7.50 | 4.92 | 1.53 | 34.43 |
| coco | text-pt | 1 | 8.05 | 8.02 | 14 | 7.00 | 4.02 | 1.74 | 42.52 |
| coco | text-ro | 1 | 8.49 | 8.47 | 14 | 7.00 | 4.24 | 1.65 | 39.36 |
| coco | text-ru | 1 | 9.70 | 9.67 | 15 | 7.50 | 4.85 | 1.55 | 35.30 |
| coco | text-sv | 1 | 8.18 | 8.15 | 15 | 7.50 | 4.09 | 1.83 | 45.48 |
| coco | text-sw | 1 | 8.63 | 8.60 | 14 | 7.00 | 4.32 | 1.62 | 38.36 |
| coco | text-te | 1 | 9.67 | 9.64 | 15 | 7.50 | 4.84 | 1.55 | 35.53 |
| coco | text-th | 1 | 8.67 | 8.64 | 13 | 6.50 | 4.33 | 1.50 | 33.31 |
| coco | text-tr | 1 | 9.05 | 9.02 | 15 | 7.50 | 4.53 | 1.66 | 39.64 |
| coco | text-uk | 1 | 9.72 | 9.68 | 15 | 7.50 | 4.86 | 1.54 | 35.23 |
| coco | text-vi | 1 | 8.15 | 8.12 | 12 | 6.00 | 4.08 | 1.47 | 32.07 |
| coco | text-zh | 1 | 8.80 | 8.77 | 14 | 7.00 | 4.40 | 1.59 | 37.15 |
| xm3600 | chameleon-512 | 1 | 11.27 | 11.24 | 12 | 6.00 | 5.63 | 1.06 | 6.09 |
| xm3600 | compvis-vq-f8-64 | 1 | 9.77 | 9.73 | 10 | 1.18 | 1.16 | 1.02 | 2.34 |
| xm3600 | compvis-vq-f8-256 | 1 | 9.69 | 9.66 | 10 | 5.00 | 4.85 | 1.03 | 3.05 |
| xm3600 | compvis-vq-imagenet-f16-1024-256 | 1 | 8.79 | 8.75 | 9 | 4.50 | 4.39 | 1.02 | 2.38 |
| xm3600 | llamagen-vq-ds16-c2i | 1 | 13.98 | 13.95 | 14 | 7.00 | 6.99 | 1.00 | 0.17 |
| xm3600 | text-ar | 1 | 10.91 | 10.89 | 14 | 0.80 | 0.62 | 1.28 | 22.05 |
| xm3600 | text-bn | 1 | 7.64 | 7.62 | 12 | 0.49 | 0.31 | 1.57 | 36.29 |
| xm3600 | text-cs | 1 | 10.13 | 10.09 | 14 | 0.66 | 0.48 | 1.38 | 27.65 |
| xm3600 | text-da | 1 | 8.76 | 8.73 | 13 | 0.85 | 0.58 | 1.48 | 32.60 |
| xm3600 | text-de | 1 | 9.31 | 9.29 | 14 | 1.44 | 0.96 | 1.50 | 33.48 |
| xm3600 | text-el | 1 | 10.32 | 10.30 | 14 | 0.80 | 0.59 | 1.36 | 26.28 |
| xm3600 | text-es | 1 | 8.53 | 8.49 | 13 | 1.14 | 0.75 | 1.52 | 34.41 |
| xm3600 | text-fa | 1 | 9.08 | 9.06 | 13 | 1.21 | 0.84 | 1.43 | 30.14 |
| xm3600 | text-fi | 1 | 11.03 | 11.01 | 14 | 0.78 | 0.62 | 1.27 | 21.18 |
| xm3600 | text-fil | 1 | 7.82 | 7.79 | 13 | 1.14 | 0.69 | 1.66 | 39.86 |
| xm3600 | text-fr | 1 | 8.44 | 8.41 | 13 | 1.51 | 0.98 | 1.54 | 35.08 |
| xm3600 | text-hi | 1 | 7.54 | 7.52 | 12 | 1.38 | 0.87 | 1.59 | 37.14 |
| xm3600 | text-hr | 1 | 10.55 | 10.53 | 14 | 0.95 | 0.72 | 1.33 | 24.62 |
| xm3600 | text-hu | 1 | 10.08 | 10.05 | 14 | 0.96 | 0.69 | 1.39 | 28.01 |
| xm3600 | text-id | 1 | 8.74 | 8.71 | 13 | 1.33 | 0.89 | 1.49 | 32.76 |
| xm3600 | text-it | 1 | 9.12 | 9.09 | 13 | 1.37 | 0.96 | 1.43 | 29.86 |
| xm3600 | text-he | 1 | 10.25 | 10.22 | 14 | 1.33 | 0.97 | 1.37 | 26.81 |
| xm3600 | text-ja | 1 | 8.48 | 8.45 | 13 | 1.44 | 0.94 | 1.53 | 34.81 |
| xm3600 | text-ko | 1 | 9.75 | 9.73 | 13 | 0.99 | 0.74 | 1.33 | 24.97 |
| xm3600 | text-mi | 1 | 7.54 | 7.51 | 12 | 0.68 | 0.43 | 1.59 | 37.16 |
| xm3600 | text-nl | 1 | 8.42 | 8.40 | 13 | 0.87 | 0.56 | 1.54 | 35.20 |
| xm3600 | text-no | 1 | 8.76 | 8.74 | 13 | 0.92 | 0.62 | 1.48 | 32.58 |
| xm3600 | text-pl | 1 | 10.27 | 10.25 | 14 | 0.90 | 0.66 | 1.36 | 26.63 |
| xm3600 | text-pt | 1 | 8.86 | 8.82 | 13 | 1.08 | 0.74 | 1.47 | 31.88 |
| xm3600 | text-ro | 1 | 9.09 | 9.06 | 14 | 1.65 | 1.07 | 1.54 | 35.07 |
| xm3600 | text-ru | 1 | 10.29 | 10.26 | 14 | 1.09 | 0.80 | 1.36 | 26.49 |
| xm3600 | text-sv | 1 | 8.72 | 8.68 | 13 | 0.82 | 0.55 | 1.49 | 32.95 |
| xm3600 | text-sw | 1 | 8.58 | 8.54 | 13 | 0.99 | 0.66 | 1.52 | 34.03 |
| xm3600 | text-te | 1 | 8.47 | 8.44 | 13 | 0.71 | 0.46 | 1.53 | 34.84 |
| xm3600 | text-th | 1 | 8.60 | 8.57 | 12 | 1.12 | 0.81 | 1.40 | 28.33 |
| xm3600 | text-tr | 1 | 10.03 | 10.00 | 14 | 0.98 | 0.70 | 1.40 | 28.37 |
| xm3600 | text-uk | 1 | 10.65 | 10.62 | 14 | 1.07 | 0.82 | 1.31 | 23.93 |
| xm3600 | text-vi | 1 | 8.68 | 8.65 | 12 | 1.61 | 1.17 | 1.38 | 27.65 |
| xm3600 | text-zh | 1 | 9.63 | 9.60 | 14 | 1.43 | 0.98 | 1.45 | 31.22 |
| spin | chameleon-512 | 1 | 11.24 | 11.20 | 12 | 6.00 | 5.62 | 1.07 | 6.37 |
| spin | compvis-vq-f8-64 | 1 | 9.75 | 9.71 | 10 | 5.00 | 4.87 | 1.03 | 2.54 |
| spin | compvis-vq-f8-256 | 1 | 9.65 | 9.62 | 10 | 5.00 | 4.83 | 1.04 | 3.49 |
| spin | compvis-vq-imagenet-f16-1024-256 | 1 | 8.77 | 8.74 | 9 | 4.50 | 4.39 | 1.03 | 2.55 |
| spin | llamagen-vq-ds16-c2i | 1 | 13.97 | 13.95 | 14 | 7.00 | 6.99 | 1.00 | 0.18 |
| cc12m | chameleon-512 | 1 | 11.28 | 11.25 | 12 | 6.00 | 5.64 | 1.06 | 6.03 |
| cc12m | compvis-vq-f8-64 | 1 | 9.76 | 9.73 | 10 | 5.00 | 4.88 | 1.02 | 2.40 |
| cc12m | compvis-vq-f8-256 | 1 | 9.64 | 9.60 | 10 | 5.00 | 4.82 | 1.04 | 3.62 |
| cc12m | compvis-vq-imagenet-f16-1024-256 | 1 | 8.73 | 8.70 | 9 | 4.50 | 4.36 | 1.03 | 3.01 |
| cc12m | llamagen-vq-ds16-c2i | 1 | 13.89 | 13.86 | 14 | 7.00 | 6.94 | 1.01 | 0.80 |
| ilsvrc | chameleon-512 | 1 | 11.27 | 11.24 | 12 | 6.00 | 5.64 | 1.06 | 6.08 |
| ilsvrc | compvis-vq-f8-64 | 1 | 9.78 | 9.74 | 10 | 5.00 | 4.89 | 1.02 | 2.21 |
| ilsvrc | compvis-vq-f8-256 | 1 | 9.69 | 9.66 | 10 | 5.00 | 4.85 | 1.03 | 3.09 |
| ilsvrc | compvis-vq-imagenet-f16-1024-256 | 1 | 8.78 | 8.75 | 9 | 4.50 | 4.39 | 1.02 | 2.40 |
| ilsvrc | llamagen-vq-ds16-c2i | 1 | 13.98 | 13.96 | 14 | 7.00 | 6.99 | 1.00 | 0.13 |
| coco | chameleon-512 | 2 | 18.80 | 18.79 | 19 | 9.50 | 9.40 | 1.01 | 1.03 |
| coco | compvis-vq-f8-64 | 2 | 18.29 | 18.28 | 19 | 9.50 | 9.14 | 1.04 | 3.74 |
| coco | compvis-vq-f8-256 | 2 | 18.12 | 18.11 | 19 | 9.50 | 9.06 | 1.05 | 4.65 |
| coco | compvis-vq-imagenet-f16-1024-256 | 2 | 17.08 | 17.05 | 18 | 9.00 | 8.54 | 1.05 | 5.13 |
| coco | llamagen-vq-ds16-c2i | 2 | 18.94 | 18.92 | 19 | 9.50 | 9.47 | 1.00 | 0.30 |
| coco | text-ar | 2 | 14.99 | 14.97 | 18 | 9.00 | 7.50 | 1.20 | 16.71 |
| coco | text-bn | 2 | 14.17 | 14.15 | 17 | 8.50 | 7.09 | 1.20 | 16.63 |

| Dataset | Model | N | ACL | E | FCL | OB | HB | CR | PR |
|---|---|---|---|---|---|---|---|---|---|
| coco | text-cs | 2 | 15.07 | 15.05 | 18 | 9.00 | 7.54 | 1.19 | 16.27 |
| coco | text-da | 2 | 12.97 | 12.95 | 17 | 8.50 | 6.49 | 1.31 | 23.70 |
| coco | text-de | 2 | 13.80 | 13.78 | 17 | 8.50 | 6.90 | 1.23 | 18.82 |
| coco | text-el | 2 | 13.46 | 13.43 | 17 | 8.50 | 6.73 | 1.26 | 20.85 |
| coco | text-es | 2 | 12.87 | 12.84 | 17 | 8.50 | 6.43 | 1.32 | 24.30 |
| coco | text-fa | 2 | 13.07 | 13.04 | 17 | 8.50 | 6.53 | 1.30 | 23.12 |
| coco | text-fi | 2 | 15.41 | 15.39 | 18 | 9.00 | 7.70 | 1.17 | 14.40 |
| coco | text-fil | 2 | 12.30 | 12.28 | 16 | 8.00 | 6.15 | 1.30 | 23.11 |
| coco | text-fr | 2 | 12.93 | 12.90 | 17 | 8.50 | 6.46 | 1.32 | 23.97 |
| coco | text-hi | 2 | 13.01 | 12.99 | 17 | 8.50 | 6.51 | 1.31 | 23.46 |
| coco | text-hr | 2 | 14.84 | 14.81 | 18 | 9.00 | 7.42 | 1.21 | 17.58 |
| coco | text-hu | 2 | 14.57 | 14.55 | 18 | 9.00 | 7.29 | 1.24 | 19.04 |
| coco | text-id | 2 | 13.28 | 13.25 | 17 | 8.50 | 6.64 | 1.28 | 21.89 |
| coco | text-it | 2 | 13.27 | 13.24 | 17 | 8.50 | 6.63 | 1.28 | 21.96 |
| coco | text-he | 2 | 15.24 | 15.22 | 18 | 9.00 | 7.62 | 1.18 | 15.31 |
| coco | text-ja | 2 | 12.08 | 12.06 | 16 | 8.00 | 6.04 | 1.32 | 24.50 |
| coco | text-ko | 2 | 13.48 | 13.46 | 17 | 8.50 | 6.74 | 1.26 | 20.69 |
| coco | text-mi | 2 | 10.90 | 10.88 | 16 | 8.00 | 5.45 | 1.47 | 31.89 |
| coco | text-nl | 2 | 13.16 | 13.13 | 17 | 8.50 | 6.58 | 1.29 | 22.59 |
| coco | text-no | 2 | 13.11 | 13.09 | 17 | 8.50 | 6.56 | 1.30 | 22.87 |
| coco | text-pl | 2 | 15.07 | 15.05 | 18 | 9.00 | 7.54 | 1.19 | 16.26 |
| coco | text-pt | 2 | 13.02 | 12.99 | 17 | 8.50 | 6.51 | 1.31 | 23.43 |
| coco | text-ro | 2 | 13.55 | 13.52 | 17 | 8.50 | 6.78 | 1.25 | 20.29 |
| coco | text-ru | 2 | 14.78 | 14.76 | 18 | 9.00 | 7.39 | 1.22 | 17.90 |
| coco | text-sv | 2 | 13.28 | 13.25 | 17 | 8.50 | 6.64 | 1.28 | 21.89 |
| coco | text-sw | 2 | 13.86 | 13.83 | 17 | 8.50 | 6.93 | 1.23 | 18.49 |
| coco | text-te | 2 | 15.02 | 15.00 | 18 | 9.00 | 7.51 | 1.20 | 16.53 |
| coco | text-th | 2 | 13.10 | 13.08 | 17 | 8.50 | 6.55 | 1.30 | 22.94 |
| coco | text-tr | 2 | 14.18 | 14.15 | 17 | 8.50 | 7.09 | 1.20 | 16.60 |
| coco | text-uk | 2 | 14.75 | 14.73 | 18 | 9.00 | 7.37 | 1.22 | 18.06 |
| coco | text-vi | 2 | 12.45 | 12.42 | 16 | 8.00 | 6.22 | 1.29 | 22.20 |
| coco | text-zh | 2 | 14.08 | 14.06 | 17 | 8.50 | 7.04 | 1.21 | 17.15 |
| xm3600 | chameleon-512 | 2 | 18.79 | 18.77 | 19 | 9.50 | 9.40 | 1.01 | 1.09 |
| xm3600 | compvis-vq-f8-64 | 2 | 16.68 | 16.63 | 17 | 1.95 | 1.91 | 1.02 | 1.87 |
| xm3600 | compvis-vq-f8-256 | 2 | 18.10 | 18.09 | 19 | 9.50 | 9.05 | 1.05 | 4.72 |
| xm3600 | compvis-vq-imagenet-f16-1024-256 | 2 | 17.04 | 17.01 | 18 | 9.00 | 8.52 | 1.06 | 5.35 |
| xm3600 | llamagen-vq-ds16-c2i | 2 | 18.94 | 18.92 | 19 | 9.50 | 9.47 | 1.00 | 0.32 |
| xm3600 | text-ar | 2 | 14.49 | 14.43 | 16 | 0.79 | 0.72 | 1.10 | 9.44 |
| xm3600 | text-bn | 2 | 11.30 | 11.27 | 14 | 0.52 | 0.42 | 1.24 | 19.31 |
| xm3600 | text-cs | 2 | 13.64 | 13.59 | 15 | 0.60 | 0.55 | 1.10 | 9.06 |
| xm3600 | text-da | 2 | 13.13 | 13.11 | 15 | 0.88 | 0.77 | 1.14 | 12.48 |
| xm3600 | text-de | 2 | 14.04 | 13.98 | 16 | 1.51 | 1.32 | 1.14 | 12.28 |
| xm3600 | text-el | 2 | 14.16 | 14.10 | 15 | 0.75 | 0.71 | 1.06 | 5.62 |
| xm3600 | text-es | 2 | 12.83 | 12.80 | 15 | 1.19 | 1.02 | 1.17 | 14.44 |
| xm3600 | text-fa | 2 | 13.66 | 13.61 | 16 | 1.37 | 1.17 | 1.17 | 14.62 |
| xm3600 | text-fi | 2 | 14.58 | 14.51 | 16 | 0.78 | 0.71 | 1.10 | 8.90 |
| xm3600 | text-fil | 2 | 12.35 | 12.32 | 15 | 1.21 | 1.00 | 1.21 | 17.64 |
| xm3600 | text-fr | 2 | 12.82 | 12.79 | 16 | 1.72 | 1.38 | 1.25 | 19.86 |
| xm3600 | text-hi | 2 | 11.30 | 11.27 | 15 | 1.60 | 1.21 | 1.33 | 24.68 |
| xm3600 | text-hr | 2 | 14.47 | 14.46 | 16 | 0.97 | 0.88 | 1.11 | 9.54 |
| xm3600 | text-hu | 2 | 14.41 | 14.39 | 16 | 0.98 | 0.88 | 1.11 | 9.96 |
| xm3600 | text-id | 2 | 12.84 | 12.80 | 15 | 1.43 | 1.22 | 1.17 | 14.40 |
| xm3600 | text-it | 2 | 13.70 | 13.65 | 16 | 1.55 | 1.33 | 1.17 | 14.35 |
| xm3600 | text-he | 2 | 14.58 | 14.52 | 16 | 1.40 | 1.28 | 1.10 | 8.88 |
| xm3600 | text-ja | 2 | 12.85 | 12.82 | 15 | 1.55 | 1.33 | 1.17 | 14.33 |
| xm3600 | text-ko | 2 | 14.00 | 13.98 | 16 | 1.09 | 0.96 | 1.14 | 12.52 |
| xm3600 | text-mi | 2 | 11.54 | 11.51 | 14 | 0.73 | 0.60 | 1.21 | 17.55 |
| xm3600 | text-nl | 2 | 12.50 | 12.48 | 15 | 0.88 | 0.73 | 1.20 | 16.67 |
| xm3600 | text-no | 2 | 12.92 | 12.89 | 15 | 0.95 | 0.82 | 1.16 | 13.87 |
| xm3600 | text-pl | 2 | 14.26 | 14.23 | 16 | 0.92 | 0.82 | 1.12 | 10.87 |
| xm3600 | text-pt | 2 | 13.40 | 13.37 | 15 | 1.14 | 1.02 | 1.12 | 10.68 |
| xm3600 | text-ro | 2 | 13.49 | 13.46 | 16 | 1.77 | 1.49 | 1.19 | 15.69 |
| xm3600 | text-ru | 2 | 14.30 | 14.28 | 16 | 1.13 | 1.01 | 1.12 | 10.63 |
| xm3600 | text-sv | 2 | 13.17 | 13.14 | 15 | 0.83 | 0.73 | 1.14 | 12.21 |
| xm3600 | text-sw | 2 | 12.98 | 12.96 | 15 | 1.04 | 0.90 | 1.16 | 13.45 |
| xm3600 | text-te | 2 | 12.11 | 12.06 | 15 | 0.71 | 0.57 | 1.24 | 19.29 |
| xm3600 | text-th | 2 | 12.63 | 12.58 | 15 | 1.30 | 1.09 | 1.19 | 15.82 |
| xm3600 | text-tr | 2 | 14.23 | 14.22 | 16 | 1.01 | 0.90 | 1.12 | 11.06 |
| xm3600 | text-uk | 2 | 14.49 | 14.48 | 16 | 1.11 | 1.01 | 1.10 | 9.41 |
| xm3600 | text-vi | 2 | 12.96 | 12.94 | 15 | 1.91 | 1.65 | 1.16 | 13.60 |
| xm3600 | text-zh | 2 | 14.31 | 14.25 | 16 | 1.52 | 1.36 | 1.12 | 10.57 |
| spin | chameleon-512 | 2 | 18.80 | 18.78 | 19 | 9.50 | 9.40 | 1.01 | 1.05 |
| spin | compvis-vq-f8-64 | 2 | 18.26 | 18.25 | 19 | 9.50 | 9.13 | 1.04 | 3.90 |
| spin | compvis-vq-f8-256 | 2 | 18.09 | 18.08 | 19 | 9.50 | 9.05 | 1.05 | 4.77 |
| spin | compvis-vq-imagenet-f16-1024-256 | 2 | 17.06 | 17.04 | 18 | 9.00 | 8.53 | 1.05 | 5.21 |
| spin | llamagen-vq-ds16-c2i | 2 | 18.94 | 18.92 | 19 | 9.50 | 9.47 | 1.00 | 0.30 |
| cc12m | chameleon-512 | 2 | 18.58 | 18.57 | 19 | 9.50 | 9.29 | 1.02 | 2.20 |
| cc12m | compvis-vq-f8-64 | 2 | 18.23 | 18.22 | 19 | 9.50 | 9.12 | 1.04 | 4.05 |
| cc12m | compvis-vq-f8-256 | 2 | 17.75 | 17.73 | 19 | 9.50 | 8.87 | 1.07 | 6.60 |
| cc12m | compvis-vq-imagenet-f16-1024-256 | 2 | 16.76 | 16.73 | 18 | 9.00 | 8.38 | 1.07 | 6.90 |
| cc12m | llamagen-vq-ds16-c2i | 2 | 18.85 | 18.83 | 19 | 9.50 | 9.42 | 1.01 | 0.81 |
| ilsvrc | chameleon-512 | 2 | 18.76 | 18.74 | 19 | 9.50 | 9.38 | 1.01 | 1.26 |
| ilsvrc | compvis-vq-f8-64 | 2 | 18.29 | 18.28 | 19 | 9.50 | 9.14 | 1.04 | 3.76 |
| ilsvrc | compvis-vq-f8-256 | 2 | 18.09 | 18.08 | 19 | 9.50 | 9.04 | 1.05 | 4.80 |
| ilsvrc | compvis-vq-imagenet-f16-1024-256 | 2 | 17.01 | 16.99 | 18 | 9.00 | 8.51 | 1.06 | 5.49 |
| ilsvrc | llamagen-vq-ds16-c2i | 2 | 18.93 | 18.91 | 19 | 9.50 | 9.47 | 1.00 | 0.36 |

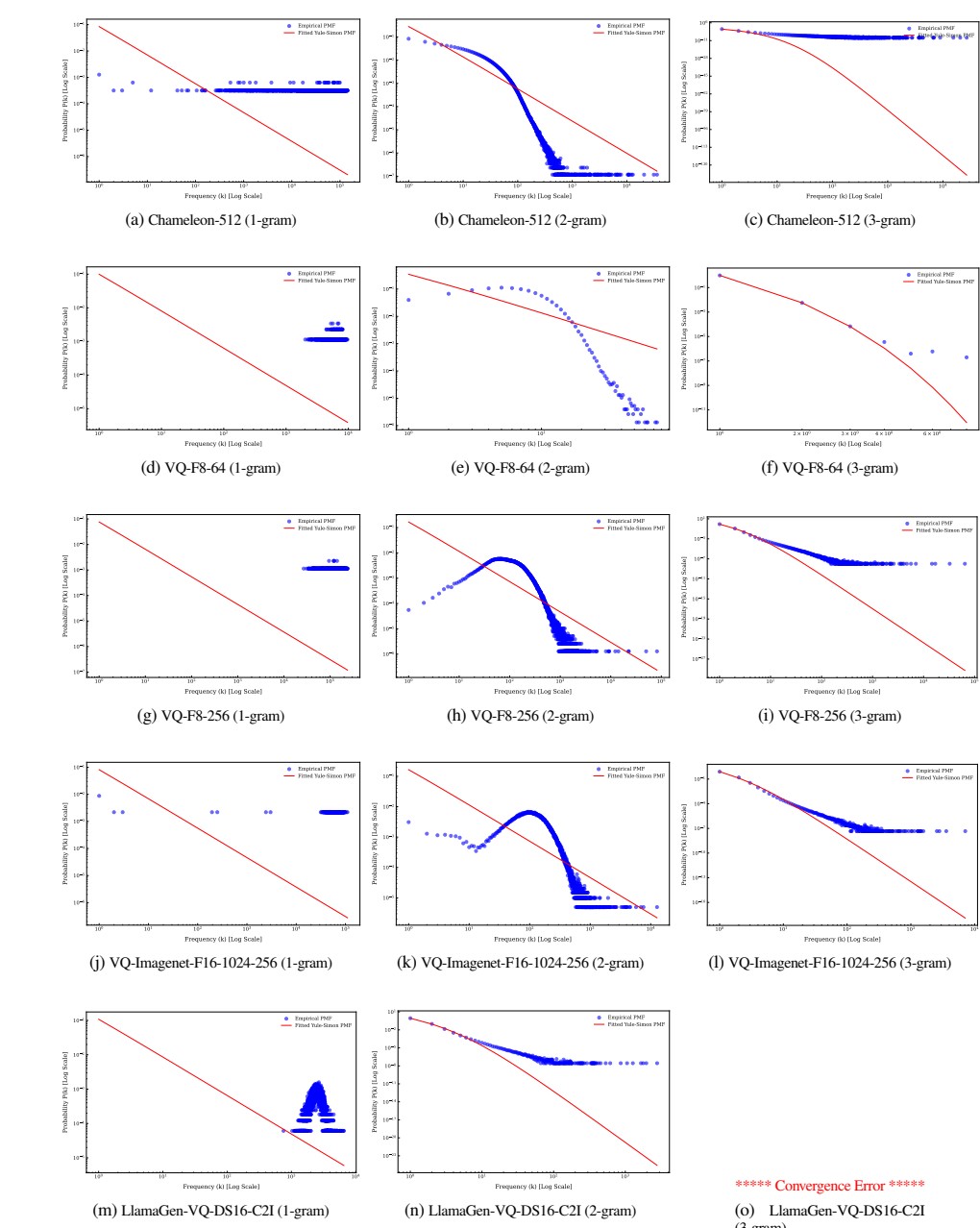

Figure F.4: N-gram analysis on various models (Simon model on XM-3600)

# I  SEGMENTATION GRANULARITY

In subsection 2.6, we explore at what level visual tokens/words correlate with parts/sub-parts/wholes of objects in images. To analyze the co-occurrence of wholes, parts, and sub-parts, we primarily leverage the SPIN dataset, discussed in Appendix B, which provides labeled annotations for each of these levels in the images. To compute co-occurrence statistics, we first extract a part-label-to-visual-token co-occurrence frequency matrix for each tokenizer and dataset. Each entry $(i,j)$ of the matrix represents the number of times that visual token $z_i$ co-occurs with part-label $y_j$ (which could represent a whole, part, or sub-part). From this co-occurrence matrix, we compute three metrics—Part Purity, Visual Token Purity, and Part-Normalized Mutual Information—as described in Hsu et al. (2021).

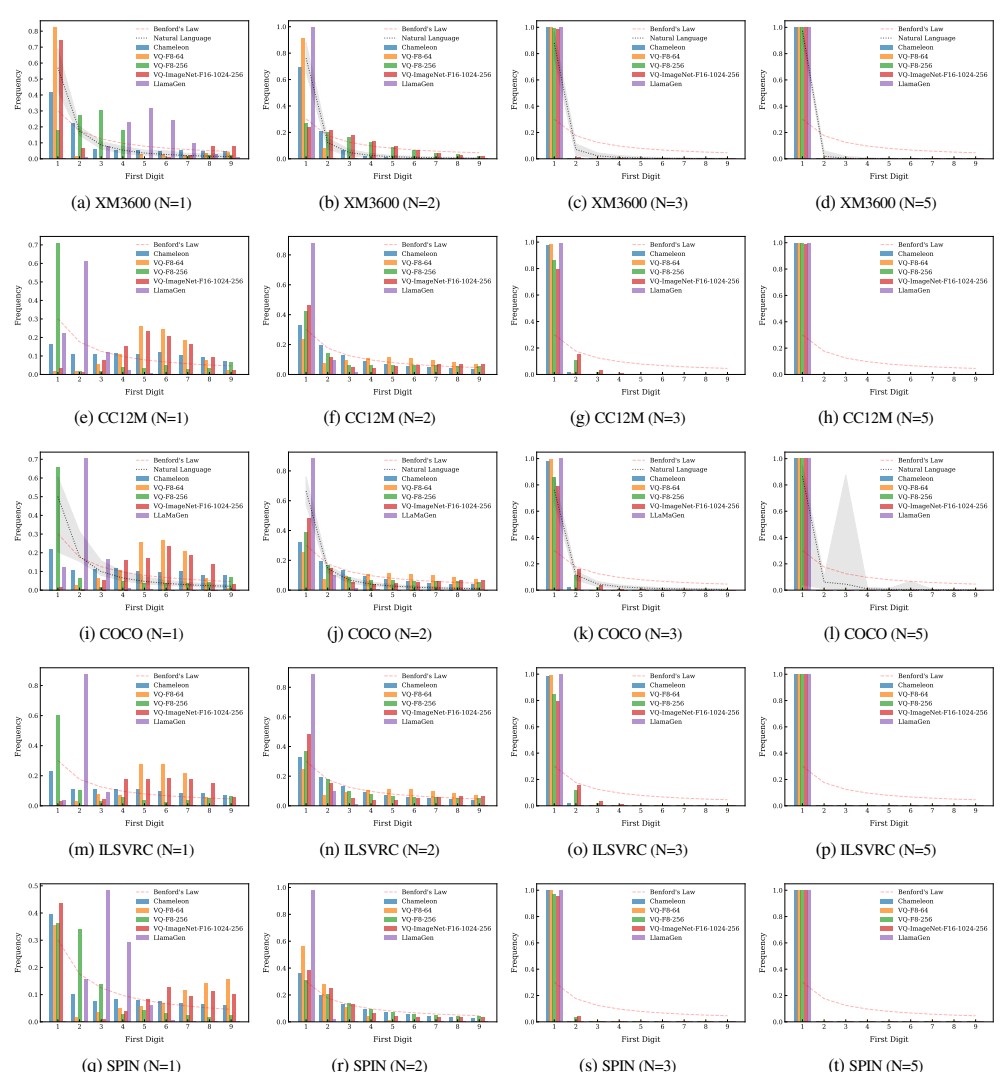

Figure G.1: Benford's Law on XM-3600, CC12M, COCO, ILSVRC, and SPIN.

**Part Purity:** Part purity describes the average probability of the most likely part-label for each visual-token, representing how accurately parts are assigned to the corresponding visual tokens. It is computed as:

$$\text{Part Purity (PP)} = \mathbb{E}_z[p(y^*(z)\,|\,z)] \tag{I.1}$$

where $z$ is a visual token cluster, $y^*(z)$ denotes the most likely part-label for a given visual-token $z$, $p(y^*(z)\,|\,z)$ is the conditional probability of the most likely part-label $y^*(z)$ given the visual-token $z$, and $\mathbb{E}_z$ is the expectation over all visual tokens. In practice, we draw these probabilities from the normalized empirical co-occurrence matrix.

**Visual Token Purity:** Visual token purity measures how well images containing the same part-label are consistently assigned to the same visual tokens. It is computed as:

$$\text{Visual Token Purity (VTP)} = \mathbb{E}_y[p(z^*(y)\,|\,y)] \tag{I.2}$$

where $y$ is a part-label, $z^*(y)$ represents the most likely visual-token for a given part-label $y$, $p(z^*(y)\,|\,y)$ is the conditional probability of the most likely visual-token $z^*(y)$ given the part-label $y$, and $\mathbb{E}_y$ is the expectation over all part-labels. Similar to part-purity, these probabilities are derived from the normalized empirical co-occurrence matrix.

**Part-Normalized Mutual Information:** Part-normalized mutual information (PNMI) is an information-theoretic metric that quantifies the percentage of uncertainty about a part-label eliminated after observing

a visual-token. It is computed as:

$$\text{PNMI} = \frac{I(y;z)}{H(y)} = \frac{H(y) - H(y \mid z)}{H(y)} = 1 - \frac{H(y \mid z)}{H(y)} \tag{I.3}$$

where $I(y;z)$ is the mutual information between part-labels $y$ and visual tokens $z$, $H(y)$ is the entropy of the part-labels, and $H(y \mid z)$ is the conditional entropy of the part-labels given the visual tokens. The entropy values are computed from the empirical co-occurrence frequency matrix, where each entry represents the joint probability $p(y,z)$ of a part-label $y$ and a visual-token $z$ co-occurring. Specifically, $H(y)$ is computed as:

$$H(y) = -\sum_i p(y_i) \log p(y_i) \tag{I.4}$$

where $p(y_i)$ is the marginal probability of part-label $y_i$, derived by summing the joint probabilities $p(y_i, z_j)$ across all visual-tokens $z_j$. Similarly, the conditional entropy $H(y \mid z)$ is computed as:

$$H(y \mid z) = -\sum_j p(z_j) \sum_i p(y_i \mid z_j) \log p(y_i \mid z_j) \tag{I.5}$$

where $p(y_i \mid z_j)$ is the conditional probability of part-label $y_i$ given visual-token $z_j$, derived from the co-occurrence matrix by normalizing the joint probabilities $p(y_i, z_j)$ by $p(z_j)$, the marginal probability of the visual-token $z_j$. Higher PNMI values indicate that more information about the part-label is captured by the visual-token assignments.

## J    TOPOLOGICAL ALIGNMENT OF VISION AND LANGUAGE TOKENS

### J.1    GLOVE EMBEDDING OF VISION AND LANGUAGE TOKENS

In order to get continuous representations of the vision and language token spaces, we employ GloVe embeddings Pennington et al. (2014). GloVe (Global Vectors for Word Representation) is a word embedding technique that captures semantic relationships between words by training on global word co-occurrence statistics. Unlike local context methods like Word2Vec (Church, 2017), GloVe constructs a matrix from word co-occurrence counts in a corpus and factorizes this matrix to generate dense vector representations. These embeddings reflect the relative meanings of words, allowing similar words to have similar vectors in the latent space. GloVe aims to learn word embeddings by factorizing a token co-occurrence matrix. The model minimizes a weighted least squares objective function:

$$J = \sum_{i,j=1}^{V} f(X_{ij}) \left( w_i^\top \tilde{w}_j + b_i + \tilde{b}_j - \log X_{ij} \right)^2 \tag{J.1}$$

where $X_{ij}$ is the co-occurrence count of token $i$ with token $j$, $w_i$ and $\tilde{w}_j$ are the token vectors for token $i$ and $j$, $b_i$ and $\tilde{b}_j$ are the bias terms, and $f(X_{ij})$ is a token co-occurrence based weighting function to discount frequent co-occurrences.

In all of the analysis methods below, before applying analysis we whiten the data before normalization to avoid significant scale effects:

$$X'_{ij} = \frac{X_{ij}}{\sigma_j}, \quad \sigma_j = \sqrt{\frac{1}{n} \sum_{i=1}^{n} (X_{ij} - \mu_j)^2}, \quad \mu_j = \frac{1}{n} \sum_{i=1}^{n} X_{ij} \tag{J.2}$$

where $X_{ij}$ is the original value of the i-th data point in the j-th feature, $\sigma_j$ is the standard deviation of the j-th feature, and $\mu_j$ is the mean of the j-th feature.

### J.2    COMPOUND PROBABILISTIC CONTEXT-FREE GRAMMARS

#### J.2.1    BACKGROUND

Here we describe the basic background and formulation of Compound Probabilistic Context-free grammars (C-PCFGs) for convenience, much of this content is sourced from (Kim et al., 2019), which we point readers to for a more thorough treatment of the topic.

C-PCFGs extend the PCFG formalism. PCFGs are defined by a 5-tuple $\mathcal{G} = (S, \mathcal{N}, \mathcal{P}, \Sigma, \mathcal{R})$, consisting of a start symbol $S$, a set of non-terminals $\mathcal{N}$, a set of pre-terminals $\mathcal{P}$, a set of terminals $\Sigma$, and a set of

derivation rules $\mathcal{R}$:

$$S \rightarrow A \qquad\qquad\qquad A \in \mathcal{N}$$
$$A \rightarrow BC \qquad\qquad\qquad A \in \mathcal{N}, B, C \in \mathcal{N} \cup \mathcal{P}$$
$$T \rightarrow w \qquad\qquad\qquad T \in \mathcal{P}, w \in \Sigma$$

The derivation rules are probabilistic, with their distribution denoted as $\boldsymbol{\pi} = \{\pi_r\}_{r \in \mathcal{R}}$. Inference may be performed efficiently over them using the inside algorithm (Baker, 1979). In neural variants of PCFGs, this distribution may be formulated as follows:

$$\pi_{S \rightarrow A} = \frac{\exp(\boldsymbol{u}_A^\top f_1(\boldsymbol{w}_S))}{\Sigma_{A' \in \mathcal{N}} \exp(\boldsymbol{u}_{A'}^\top f_1(\boldsymbol{w}_S))}$$

$$\pi_{A \rightarrow BC} = \frac{\exp(\boldsymbol{u}_{BC}^\top \boldsymbol{w}_A)}{\Sigma_{B'C' \in \mathcal{M}} \exp(\boldsymbol{u}_{B'C'}^\top \boldsymbol{w}_A)}$$

$$\pi_{T \rightarrow w} = \frac{\exp(\boldsymbol{u}_w^\top f_2(\boldsymbol{w}_T))}{\Sigma_{w' \in \Sigma} \exp(\boldsymbol{u}_{w'}^\top f_2(\boldsymbol{w}_T))}$$

where $\boldsymbol{u}$ are transformation vectors for each production rule, $\boldsymbol{w}$ are learnable parameter vectors for each symbol, and $f_1$ and $f_2$ are neural networks.

Compound PCFGs (Kim et al., 2019) formulate rule probabilities as a compound probability distribution (Robbins, 1956):

$$\boldsymbol{z} \sim p_\gamma(\boldsymbol{z}) \qquad\qquad\qquad \boldsymbol{\pi}_{\boldsymbol{z}} = f_\lambda(\boldsymbol{z}, \boldsymbol{E}_\mathcal{G})$$

Where $\boldsymbol{z}$ is a latent variable generated by a prior distribution (a spherical Gaussian) and $\boldsymbol{E}_\mathcal{G} = \{\boldsymbol{w}_N | N \in \{S\} \cup \mathcal{N} \cup \mathcal{P}\}$ denotes the set of symbol embeddings. Rule probabilities $\boldsymbol{\pi}_{\boldsymbol{z}}$ are conditioned on this latent:

$$\pi_{\boldsymbol{z}, S \rightarrow A} \propto \exp(\boldsymbol{u}_A^\top f_1([\boldsymbol{w}_S; \boldsymbol{z}])),$$

$$\pi_{\boldsymbol{z}, A \rightarrow BC} \propto \exp(\boldsymbol{u}_{BC}^\top [\boldsymbol{w}_A; \boldsymbol{z}]),$$

$$\pi_{\boldsymbol{z}, T \rightarrow w} \propto \exp(\boldsymbol{u}_w^\top f_2([\boldsymbol{w}_T; \boldsymbol{z}]))$$

The latent $\boldsymbol{z}$ allows global information to be shared across parsing decisions, while simultaneously respecting the context-free assumption when $\boldsymbol{z}$ is fixed, allowing for efficient inference as before.

C-PCFGs are optimized with variational methods (Kingma, 2013), since the introduction of $\boldsymbol{z}$ makes inference intractable. At inference time, given a sentence $\boldsymbol{x}$, the variational inference network $q_\phi$ is used to produce the latent $\boldsymbol{z} = \boldsymbol{\mu}_\phi(g(\mathcal{E}(\boldsymbol{x})))$. Here, $g$ is a sentence encoder used to generate a vector representation given token embeddings $\mathcal{E}(\boldsymbol{x})$. For more details on C-PCFGs, we point readers to Kim et al. (2019).

### J.2.2 PARSE TREES

In Figure J.1 we show an example parse tree generated with a learned grammar for each dataset.

### J.3 PROCRUSTES ANALYSIS

Procrustes Analysis is a statistical method used to compare the shapes or structures of two datasets by finding an optimal transformation (including translation, scaling, and rotation) that minimizes the distance between corresponding points in the datasets. The resulting transformation provides insight into how closely the datasets align in their geometry. Procrustes Analysis minimizes the distance between two matrices $X$ and $Y$ by finding the optimal translation, scaling, and rotation. The goal is to solve:

$$\min_{R, b, c} \|bXR + c - Y\|_F \tag{J.3}$$

where: $X$ and $Y$ are the two point sets (matrices) being compared, $R$ is the optimal rotation matrix, $b$ is a scaling factor, $c$ is the translation vector, and $\|\cdot\|_F$ is the Frobenius norm.

For Procrustes analysis, it is required that the two matrices to be aligned have identical shape. Because the number of tokens is different in the vision and language cases, in our experiments we use K-means to quantize the different token embedding spaces to 256 centers, which we then compare topologically. This has the downside of reducing the topological comparisons to more global structure comparisons, however means that we can run experiments on point-to-point coherence.

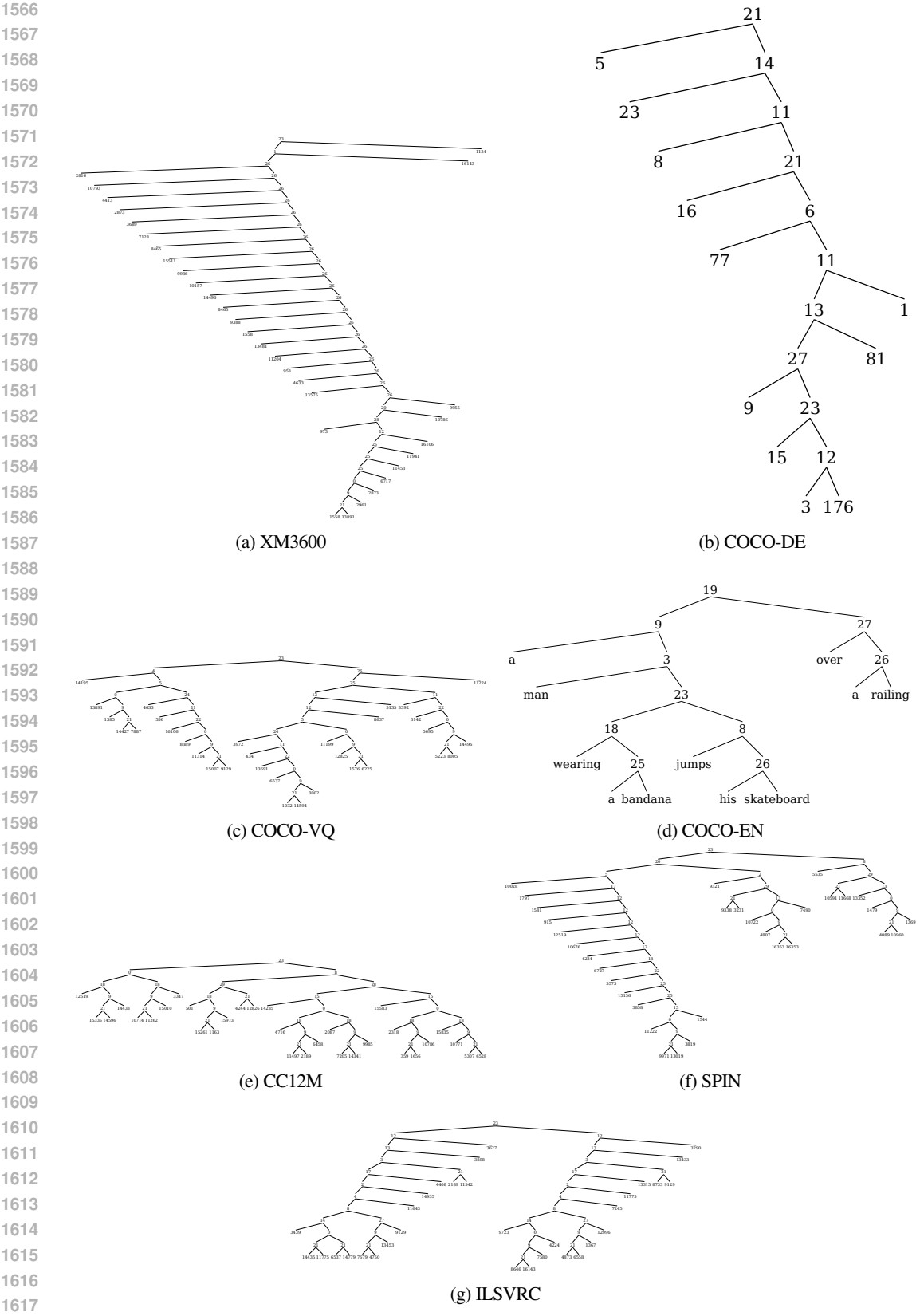

Figure J.1: Example parse trees for different datasets.

Full results for our Procrustes analysis are given in Figure J.2. For simplicity and clarity, in Figure J.2, we replace the distance with $1-$distance to get a similarity measure, and set the diagonal to 0 (even though the diagonal similarity is naturally 1), in order to avoid contrast issues on the off-diagonals.

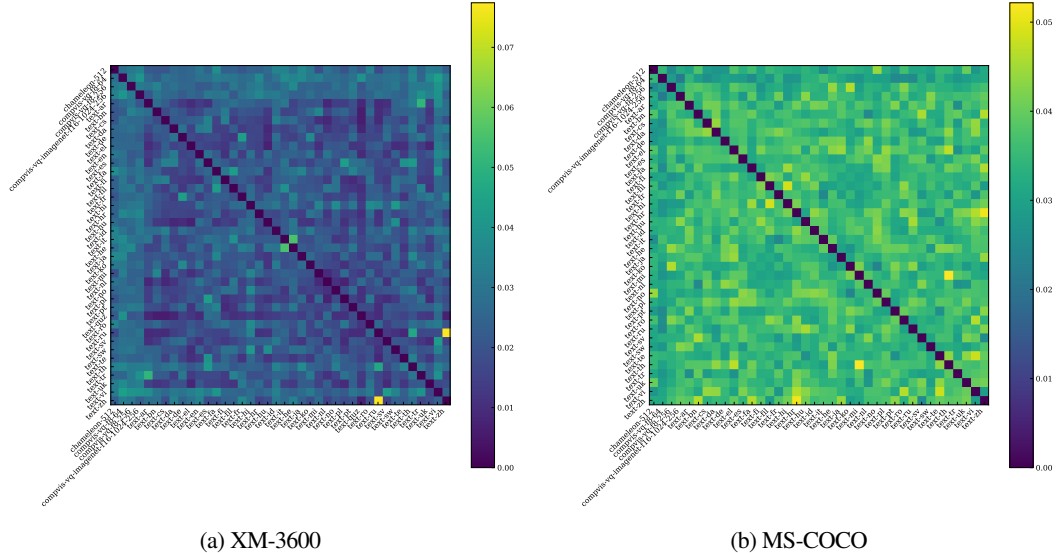

(a) XM-3600

(b) MS-COCO

Figure J.2: Procrustes similarity matrices for XM-3600 and MS-COCO across all models and languages.

### J.4  DIRECTED HAUSDORFF DISTANCE

Directed Hausdorff Distance is a measure used to quantify the degree of mismatch between two point sets by calculating the greatest distance from a point in one set to the nearest point in the other set. It considers only the largest such deviation in one direction, making it useful for determining the extent to which one set is contained within or approximates another. The directed Hausdorff distance from set $A$ to set $B$ is defined as:

$$d_H(A,B) = \max_{a \in A} \min_{b \in B} \|a-b\| \tag{J.4}$$

where $A$ and $B$ are two point sets, and $\|a-b\|$ is the Euclidean distance between point $a \in A$ and point $b \in B$.

Similar to Figure J.2, for simplicity and clarity, in Figure J.3 we replace the distance with $1-$distance to get a similarity measure, and set the diagonal to 0 (even though the diagonal similarity is naturally 1), in order to avoid contrast issues on the off-diagonals.

Full results for the directed Haussdorf distance are given in Figure J.3.

## K  IMPLICATIONS

In conclusion, we hope that this paper demonstrates the following enumerated findings and implications, demonstrated in Table K.1:

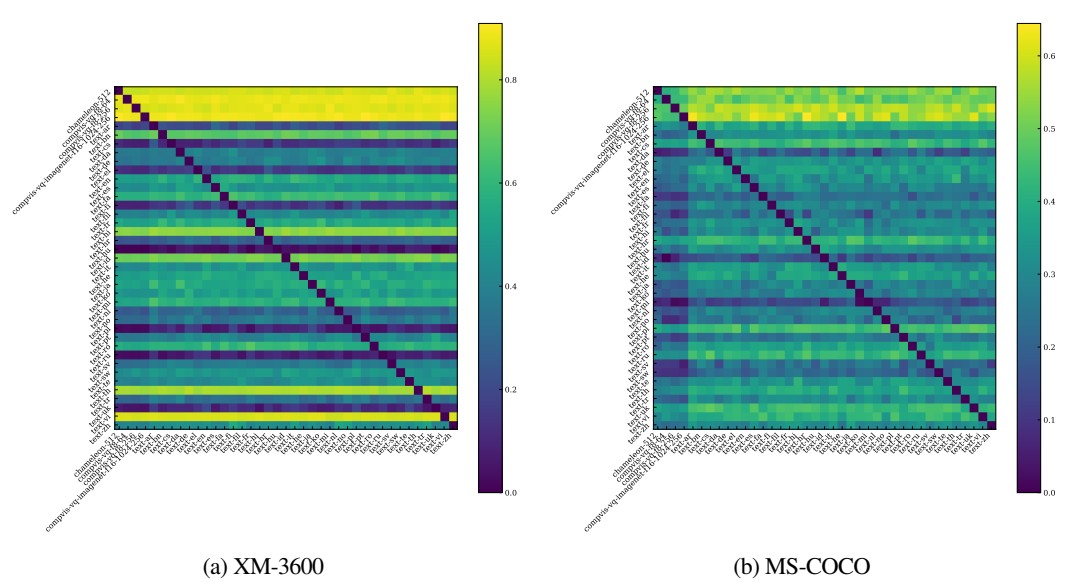

(a) XM-3600          (b) MS-COCO

Figure J.3: Directed Haussdorf *similarity* matrices for XM-3600 and MS-COCO across all models and languages.

| Topic | Description | Implications |
|---|---|---|
| **Statistical Properties of Visual Tokens** - subsection 2.2 | Visual tokens follow a Zipfian distribution but deviate significantly from natural language variants, exhibiting greater per-token entropy and lower compressibility than natural language tokens. | Models for visual languages may require larger embeddings, more attention heads, and additional training time to handle this complexity. |
| **Higher Token Innovation** - subsection 2.3 | New images introduce a rapid increase in unique tokens, significantly higher than what is observed in natural languages. | Models should handle high vocabulary diversity to avoid overfitting and may require even more visually diverse datasets than natural language applications for effective training. |
| **Low Compressibility and High Complexity** - subsection 2.5 | Visual languages show higher entropy and low compressibility, with distributed and complex token relationships. | Deeper, denser models with significant representation capacity are necessary to capture the relationships and hierarchical structures in visual tokens. |
| **Granularity and Representation** - subsection 2.6 | Visual tokens represent intermediate granularity, capturing parts of objects rather than entire objects or fine details. | Models may need to prioritize mid-level representations, as tokenizers align more with object parts than whole-object structures. |
| **Weaker Grammatical Structures** - section 3 | Visual languages lack cohesive grammatical structures, leading to high perplexity and fragmented grammar rules compared to natural languages. | Grammar-based models may be less effective for visual languages, and alternative structural representations could improve performance. |
| **Model-Specific Alignment** - subsection 3.1 | Visual token representations vary significantly across models and are poorly aligned with natural languages. | Future architectures should aim to reduce alignment asymmetry between textual and visual spaces, possibly through shared embeddings or improved tokenization. |

Table K.1: Key observations and implications for vision-language modeling discussed in this paper.

