# OpenReview forum: "Analyzing the Language of Visual Tokens"
_ICLR.cc/2025/Conference — Submitted to ICLR 2025_

### Official Review · Reviewer_UMeH · 2024-10-25

**Soundness:** 3
**Presentation:** 4
**Contribution:** 3
**Rating:** 8
**Confidence:** 3

**Summary:**

The paper studies the language expressed through “visual tokens” – discrete tokens representing images, that are obtained from an image tokenizer followed by linearization. Such tokens are used jointly with language tokens in multi-modal transformer-based models, such as Chameleon. The paper analyzes various statistical properties of the language induced by visual tokens, such as its entropy, naturality, and topology, in comparison to natural human language and language generated by language models. The paper shows multiple interesting findings highlighting similarities and differences between “visual languages” and natural languages. The authors also include several discussions regarding the potential implications of their findings, suggesting ways in which they can inform the design of more effective vision models.

**Strengths:**

* Multi-modal models, especially those involving vision and language, are of growing usage and interest. The paper takes a unique stance on analyzing these models, viewing visual tokens as language and applying well-established empirical statistics methods.

* I found the analysis very comprehensive, covering 5 different tokenizers, multiple datasets, and more than 6 methods. It also tackles fundamental research questions.

* The paper is clearly written and easy to read, balancing well between concrete results and intuitive interpretations.

**Weaknesses:**

* The main drawback of the paper is that while it discusses multiple potential implications of the reported findings, it does not demonstrate them, leaving practical exploration of these implications for future work. I find the observations interesting and important by themselves, but the paper could be made much stronger with such experiments.

* I may have overlooked it, but there is very little discussion on the differences across visual tokenizers and datasets. If it implies that current tokenizers largely behave similarly (regardless of hyperparameters, etc) then this should be emphasized.

* Not a major weakness, but a point for consideration is that the paper focuses on *discrete* visual tokens, while recent performant models like LLaVA rely on *continuous* tokens. The authors acknowledge that, noting that some of the analyses could be applied to continuous tokens, but they leave the exploration of this for future work. However, I think the paper could be made more impactful by applying its unique analysis to continuous tokens as well and comparing them to the discrete tokens.

**Questions:**

Typos:
* L73 – “we aim to show through these experiments show that”
* L79 – “The first, question that we examine“

---

> ### Author Response · Authors · 2024-11-20
>
> We greatly appreciate the reviewer for taking the time to review our manuscript and provide thoughtful comments, and we appreciate that the reviewer highlights our comprehensive analysis, unique stance on fundamental research questions, and clear presentation. We would like to take this opportunity to address some of the reviewer’s concerns:
>
> **Follow-up experiments (W1):**  We acknowledge that additional follow-up experimental validation would enhance the impact of our results – indeed, demonstrating that even one of these effects is significant would itself be a significant contribution. To the extent possible, we have cited where the identified implications/conclusions have **already** been validated with existing experimental results (such as in Section 2.5, where Tong et al. [1] validate empirically that an increase in transformer depth can improve performance), and such pre-hoc experimental confirmation (we hope) gives a basis for confidence that our method of analysis can be leveraged to improve existing approaches. Unfortunately, such validations can be both extremely compute intensive, and subject to high variance, and we believe that each experimental confirmation itself represents a large contribution beyond the existing analysis and design methodology, which we believe is the key contribution of this work. We appreciate that the reviewer recognizes that even despite limited experimental exploration of some proposed contributions, the observations themselves are interesting and important.
>
> **Differences between tokenizers and datasets (W2):** While we do discuss the tokenizers themselves in Appendix A and datasets in appendix B, we agree that a more comprehensive discussion of the fine-grained differences between the tokenizers/datasets would be valuable. We have added additional discussion to the appendix (A/B), and will highlight some of the universality of these findings in the paper.
>
> **Discrete vs. Continuous Tokenization (W3):** We strongly agree that it would be valuable to explore additional tokenization strategies, particularly hybrid or continuous tokenizers (such as LLaVA). That being said, such an extension is highly non-trivial: because the natural language techniques that we explore in this paper were developed for discrete tokens (such as are common in natural language), each would have to be extended to the continuous domain.
>
> For some of these analyses, such as entropy, continuous domain generalizations exist (such as differential entropy), however are challenging to quantify in higher dimensional spaces, and to our knowledge, it would be foundational work in probability theory to demonstrate that such entropy values are comparable to those in the discrete domain.  For other analyses, such as Benford’s law, no such continuous domain generalization exists, and would have to be derived from first-principles. While it appears that there is some intuition as to the underlying foundational principles behind Benford’s law [1], we believe that simply deriving (and demonstrating) such a continuous generalization would be a significant contribution in several communities. Similar techniques would have to be derived for other methods such Yule-Simon laws or C-PCFGs, each of which we believe would be significant individual contributions.
>
> It is possible to perform analyses on quantized spaces of the continuous domain, treating the quantized states as continuous variables, however doing so introduces significant quantization bias that can impact the outcomes. For example when analyzing entropy in quantized spaces, the resolution of quantization directly impacts the calculated entropy. Coarse quantization tends to underestimate the entropy by failing to capture the full variability of the continuous domain, while fine quantization can overfit noise in the data. Similarly, for Yule-Simon distributions, the observed frequencies of quantized states would have the potential to reflect artifacts of binning rather than true reflections of the underlying continuous distribution. Thus, the resulting power-law exponent might be systematically distorted, either attenuated or exaggerated, based on the quantization scheme used.
>
> Given these considerations, we decided to omit continuous valued tokenizers from this work, as there remain a significant number of unanswered questions that would have to be addressed (either at a theoretical level, or at a quantization-validation level), however we strongly believe that future work can make strides towards evaluating such rules in the continuous domain as well. We have further clarified this in appendix C, to provide more context for this decision.
>
>
> **Other Notes:** We have addressed the typos mentioned by the reviewer in the latest revision.
>
> [1] Shengbang Tong, et al. Cambrian-1: A fully open, vision-centric exploration of multimodal llms

---

> > ### Comment · Reviewer_UMeH · 2024-11-25
> >
> > Thanks for the additional clarifications and discussion!
> >
> > Reviewer UMeH

---

### Official Review · Reviewer_1eu7 · 2024-11-01

**Soundness:** 2
**Presentation:** 3
**Contribution:** 2
**Rating:** 3
**Confidence:** 3

**Summary:**

This paper tries to investigate the similarity between language tokens and visual tokens. More specifically, the authors inspect the equivalence of visual and natural languages through an empirical analysis of token distributions, segmentation granularity, and syntactic and semantic structures.

**Strengths:**

1. By examining visual tokens through the lens of linguistic principles, the authors provide a novel framework for analyzing multimodal models, which enriches the discourse surrounding the integration of vision and language.
2. The comparison of visual languages to natural languages, especially in terms of grammatical structure and co-occurrence patterns, yields interesting conclusions about the nature of visual representation and its implications for model design.
3. The paper writing is clear. This clarity aids in understanding the complex interactions between visual and linguistic modalities.

**Weaknesses:**

1. The paper primarily borrows conclusions and formulas from the realm of language models and applies them to visual tokens without offering substantial new insights specific to visual representations. The analysis and discussion often appear superficial, failing to yield novel conclusions regarding the unique characteristics of visual tokens.
2. The study lacks original analytical frameworks or targeted statistical experiments designed specifically for visual tokens. As a result, the manuscript reads more like an experimental record rather than a comprehensive exploration that provides meaningful inspiration or deeper understanding for the reader.
3. The conclusions drawn regarding model design and their implications for improving multimodal task performance are notably unclear. The recommendations do not provide concrete guidance on how to effectively implement these insights in practical model development, leaving the reader uncertain about their applicability in real-world scenarios.

**Questions:**

1. How does your analysis of visual tokens extend beyond existing conclusions from the NLP domain? What specific novel insights can you offer regarding the unique properties of visual tokens?
2. What additional analyses or experiments could be conducted to deepen the understanding of visual tokens? Are there specific hypotheses or exploratory questions that you believe warrant further investigation?
3. Can you clarify how the conclusions drawn in your paper can be practically applied to improve the design of multimodal models? What specific recommendations do you have for researchers looking to leverage your findings in their model development?
4. In your comparisons of visual languages and natural languages, what specific aspects do you believe are most critical to consider when developing models that integrate both modalities? How do these aspects influence the training strategies employed?

---

> ### Author Response · Authors · 2024-11-20
>
> We greatly appreciate the reviewer for taking the time to review our manuscript and provide thoughtful comments, and we appreciate that the reviewer highlights our clear presentation, novel framework for analyzing multimodal models, and interesting conclusions about the nature of visual representation and its implications for model design.  We would like to take this opportunity to address some of the reviewer’s  questions and concerns:
>
> **Use of Language-Based Analysis for Vision-Based Models (W1):** We strongly believe that current existing research lacks an in-depth understanding of whether the internal structure of visual tokens mirrors the principles governing natural languages. Because existing models for vision and language (such as LLaVA) treat vision as 1-D sequence learning with next token prediction, we explore in this paper if it is statistically reasonable to do so. Specifically, we ask the question: do languages formed of visual tokens follow the same statistical patterns, such as frequency distributions, grammatical rules, or semantic dependencies, that human languages exhibit?
>
> To answer these questions, we apply existing statistical analyses from NLP, including zipf’s law, yule-simon’s law for token innovation, Benford’s law for naturalness, measures of entropy and compressibility, measures of object granularity, applications of grammatical structure, and analyses of topological alignment. These approaches allow us to analyze key distributional differences between vision and natural language token sequences when such token sequences are treated as 1-D natural language sequences (as they are in modern transformer architectures).
>
> Thus, we believe that it is **not a weakness, but a strength**, that we rely on existing statistical analyses, as they allow us to compare the applicability of techniques designed for NLP to vision models.
>
> **Lack of New Analytical Frameworks for Visual Tokens (W2):** While we apply NLP tools, our work is the first to explore their application to visual token analysis, uncovering previously unknown phenomena such as high token innovation rates, non-linear N-gram distributions, and fragmented grammatical structures. These findings provide the first evidence that visual tokens diverge significantly from natural language sequences, challenging existing assumptions in multimodal modeling.
>
> In principle, we agree with the reviewer that new analytical frameworks _could_ be developed to analyze visual tokens, however, in this paper we take a slightly different approach and aim to answer the question: “Do **existing** tools in natural language processing provide insights into if/how sequences of vision tokens differ from their natural language counterparts?” By demonstrating these unique statistical properties, our work challenges the prevailing assumption that methods developed for natural language processing can be seamlessly applied to visual token sequences, a conclusion that has important implications for how multimodal models are designed and evaluated (which we discuss below).

---

> ### Author Response · Authors · 2024-11-20
>
> **No Concrete Insights (W3), Can you clarify how the conclusions drawn in your paper can be practically applied to improve the design of multimodal models? (Q3.1), What specific recommendations do you have for researchers looking to leverage your findings in their model development? (Q3.2)**:
>
> While our work is, indeed, primarily analytical, we would like to emphasize the following concrete insights into model architecture and design that result from the experiments in our paper:
>
> 1. **Statistical Properties of Visual Tokens** (Section: 2.2 - Token Frequency and Zipf’s Law)
>    - Visual tokens follow a Zipfian distribution but deviate significantly, exhibiting greater per-token entropy and lower compressibility than natural language tokens.
>    - Implications: Models for visual languages may require larger embeddings, more attention heads, and additional training time to handle this complexity.
>
> 2. **Higher Token Innovation** (Section: 2.3 - Token Innovation)
>    - New images introduce a rapid increase in unique tokens, significantly higher than what is observed in natural languages.
>    - Implications: Models should handle high vocabulary diversity to avoid overfitting and may require even more visually diverse datasets than natural language applications for effective training.
>
>
> 3. **Low Compressibility and High Complexity** (Section: 2.5 - Entropy and Redundancy)
>    - Visual languages show higher entropy and low compressibility, with distributed and complex token relationships.
>    - Implications: Deeper, denseer models with significant representation capacity are necessary to capture the relationships and hierarchical structures in visual tokens.
>
> 4. **Granularity and Representation** (Section: 2.6 - Token Segmentation Granularity)
>    - Visual tokens represent intermediate granularity, capturing parts of objects rather than entire objects or fine details.
>    - Implications: Models may need to prioritize mid-level representations, as tokenizers align more with object parts than whole-object structures.
>
> 5. **Weaker Grammatical Structures** (Section: 3 - Are Visual Languages Structured Like Natural Languages?)
>    - Visual languages’ structure cannot be adequately modeled  using standard grammatical structures (context-free grammars) used successfully in NLP, as indicated by  high perplexity and fragmented grammar rules compared to natural languages.
>    - Implications: Linearization may be throwing away important structural information for visual languages , suggesting that  alternative structural representations could improve performance.
>
> 6. **Model-Specific Alignment** (Section: 3.1 - Topological Similarity)
>    - Visual token representations vary significantly across models and are poorly aligned with natural languages.
>    - Implications: Future architectures should aim to reduce alignment asymmetry between textual and visual spaces, possibly through shared embeddings or improved tokenization.
>
> We strongly believe that this collection of observations, paired with the comprehensive analysis in the paper and concrete statistical motivations behind these potential differences provide a clear basis for further research into the design of LLM-based multimodal models. To address this in the paper, we have clearly enumerated these findings in Section 4 and the new appendix, Appendix K.
>
> **How does your analysis of visual tokens extend beyond existing conclusions from the NLP domain? (Q1.1), What specific novel insights can you offer regarding the unique properties of visual tokens? (Q1.2):**
>
> We would like to strongly emphasize that our work is the first and only approach to explore the statistical foundations behind sequences of visual tokens - and no existing work in NLP or Vision has explored the underlying statistics of sequences of vision tokens generated by modern tokenizers used in the context of vision-language transformers.  Our analysis in this clearly demonstrates several unique properties of visual tokens that diverge from the structured rules of natural language (similar to those claims in W3): Visual tokens, unlike natural languages, show higher entropy (Section 2.5), greater token innovation (Section 2.3), and lack cohesive grammatical structures (Section 3). This complexity, combined with their fragmented representation (Section 2.6) and non-linear token distributions (Section 2.2), highlights why NLP-inspired methods like grammar-based modeling or compression algorithms are insufficient alone. Overall, these findings demonstrate that modality-specific approaches are necessary to effectively capture the unique nature of visual languages. We have clearly enumerated these findings in Section 4 and the new appendix, Appendix K.

---

> ### Author Response · Authors · 2024-11-20
>
> **What additional analyses or experiments could be conducted to deepen the understanding of visual tokens? (Q2.1):** We strongly believe that in this work, we have demonstrated a large number of insights and innovations regarding visual tokens (See the answer to W3). We are happy to consider additional specific analyses in the final version if the reviewer has any concrete suggestions.
>
> **Are there specific hypotheses or exploratory questions that you believe warrant further investigation? (Q2.2)**:  Throughout the paper we suggest several directions for interesting future research, including a detailed discussion in Appendix C of our limitations and directions for future work. Key avenues for future research include: (1) extending our analysis to continuous tokenization methods to validate their applicability, (2) examining the impact of dataset diversity on token innovation, and (3) designing new tokenization or decoding strategies optimized for the high entropy and granularity of visual tokens. Enumerated, some of these questions are:
>
> - Do similar artifacts exist in continuous tokenization approaches? (Overall, Appendix C)
> - What is the impact of dataset diversity and size on token innovation and the quality of generated embeddings? (Section: 2.3, Appendix C)
> - How does the scan order used in tokenization impact the compressibility and encoding efficiency of visual tokens? (Section: 2.5, Appendix C)
> - What modifications to decoding strategies, such as frequency penalties, can improve inference for visual tokens? (Section: 2.5)
> - Are there more effective ways to model the intermediate granularity of visual tokens (e.g., object parts) to improve multimodal performance? (Section: 2.6, Appendix C)
> - How does the lack of cohesive grammatical structures in visual tokens affect their usability in grammar-based modeling approaches? (Section: 3)
> - Can new grammatical formalisms (e.g., mildly context-sensitive grammars) better capture the structures of visual tokens compared to context-free grammars? (Section: 3)
> - Are there better methods to tokenize visual data that improve alignment with natural language embeddings? (Section: 3.1)
> - How can model architectures be optimized to reduce the asymmetry between visual and textual latent spaces? (Section: 3.1)
> - Are there ways to align certain languages more effectively with visual tokens, considering observed cross-linguistic variations in alignment strength? (Section: 3.1)
> - Does the behavior of visual tokens change in video data (non-static images)? (Appendix C)
>
> We are happy to explore additional avenues for further research if the reviewer has any suggestions.
>
> **In your comparisons of visual languages and natural languages, what specific aspects do you believe are most critical to consider when developing models that integrate both modalities? (Q4.1) How do these aspects influence the training strategies employed? (Q4.2)**:
>
> The key contribution of our work, as discussed in Section 4, and through the paper, is that when developing models that integrate visual and natural languages, it is critical to consider the fundamental differences in token distributions, as natural language token distributions clearly differ from those of visual tokens. This has wide reaching effects (as we discuss in the response to W3), and we have clarified these effects in Section 1 of the paper, as well as in Section 4/Appendix K.

---

> ### Author Response · Authors · 2024-11-27
>
> We appreciate the time and effort you've invested in reviewing our work! In our rebuttal and the updated paper, we hope that we've thoroughly addressed the concerns and suggestions raised in the initial reviews. We are happy to discuss any of these points further, and we are eager to engage in more discussion during the official ICLR discussion period. If you find that all your questions have been resolved, we kindly ask you to consider reflecting this in the scores.
>
> Thank you once again for your thoughtful feedback and consideration!
>
> Best regards,
> The Authors

---

### Official Review · Reviewer_oK1p · 2024-11-03

**Soundness:** 2
**Presentation:** 3
**Contribution:** 2
**Rating:** 5
**Confidence:** 3

**Summary:**

The paper analyzes the discrete visual tokens learned with VQ-VAEs and investigates several of their properties, such as their frequency distributions, grammatical structures, and topological alignments with natural languages. Some of their findings include: 1) 1-gram and 2-gram visual tokens do not follow Zipf's law, but longer n-grams match it better. 2) visual tokenizers are good at capturing part-level representations. 3) Context-free grammar may not as accurately represent "visual languages" as natural languages. 4) "vision languages" align less with each other than with natural languages.

**Strengths:**

1. The idea of analyzing the visual tokens learned by VQ-VAE models and investigating their properties using natural language tools is interesting and could be inspiring.
2. The writing is clear, with each research question and findings clearly stated in each of the sections.

**Weaknesses:**

1. The paper presents various findings and suggests potential implications and directions for future work. However, it lacks follow-up experiments to support these hypotheses, making the claims less convincing and raising doubts about the practical value of the findings.
2. The approach of applying language tools directly to visual tokens is questionable, especially as the findings suggest that visual tokens may not inherently exhibit natural language structures. Additionally, it remains uncertain whether applying 1D-based methods to 2D visual tokens is appropriate.
3. Overall, it seems to me that the results presented do not convincingly demonstrate that using language tools on visual tokens is a valid method for analysis or clarifying the implications of their similarity or dissimilarity to natural language structures. The paper provides broad analyses, but a deeper focus on fewer points with more experiments would strengthen the claims.

**Questions:**

1. How do you define n-grams for visual tokens obtained from 2D images?
2. Is it possible that "vision languages" actually follow natural language properties, but require vision-specific analysis tools or adaptations of existing tools to reveal them?

---

> ### Author Response · Authors · 2024-11-20
>
> We greatly appreciate the reviewer for taking the time to review our manuscript and provide thoughtful comments, and we appreciate that the reviewer highlights our clear writing and novel approach. We would like to take this opportunity to address some of the reviewer’s  questions and concerns:
>
> **Follow-up experiments (W1):**  We acknowledge that additional follow-up experimental validation for some experiments would significantly enhance the impact of our results – indeed, demonstrating that _even one of these effects is significant would itself be a significant contribution_. That being said, we hope that, as mentioned by reviewer UMeH, we have demonstrated the observations themselves are interesting and important.
>
> To address this, to the extent possible, we have cited where the identified implications/conclusions have **already** been validated with existing experimental results (such as in Section 2.5, where Tong et al. [1] validate empirically that an increase in transformer depth can improve performance), and such pre-hoc experimental confirmation (we hope) gives a basis for confidence that our method of analysis can be leveraged to improve existing approaches. Unfortunately, such validations can be both extremely compute intensive, and subject to high variance, and we believe that each experimental confirmation itself represents a large contribution beyond the existing analysis and design methodology, which is the key contribution of this work.
>
> **Using language-tools for vision (W2, Q2):** We strongly agree with the reviewer that using 1-D methodology for 2-D tasks is often insufficient, and indeed, more broadly, that using architectures designed for 1-D tasks is likely to lead to suboptimal performance compared to using model architectures designed from the ground up for 2D vision learning. Unfortunately, many existing model architectures for multimodal learning (such as Chameleon, or LLaVA) rely on techniques that are spatially unaware (such as RoPE, a position encoding technique which doesn’t use any 2D information), or architectures designed for natural language applications (like few-head multi-head attention).
>
> In this work, we aim to demonstrate the _unique nature of 2D data_ by looking at precisely how 2D visual-token distributions differ from their 1D natural language counterparts. We look at these differences primarily through the lens of existing statistical analyses from NLP, including zipf’s law, yule-simon’s law for token innovation, Benford’s law for naturalness, measures of entropy and compressibility, measures of object granularity, applications of grammatical structure, and analyses of topological alignment. Overall, these experiments allow us to analyze key distributional differences between vision and natural language token sequences when such token sequences are treated as 1-D natural language sequences (as they are in modern transformer architectures). We believe that the fact that visual tokens may not inherently exhibit natural language structures is itself an interesting and relevant contribution - as reviewer cqku mentions - and we believe that the results of our analysis significantly motivate vision-specific architectures compared to using the same architectures that work for 1-D natural language data distributions.
>
> Thus, overall, we believe that it is **not a weakness, but a strength**, that we rely on existing 1-D statistical analyses, as they allow us to compare the applicability of techniques designed for NLP to vision models. We have added additional wording in the introduction to clarify this point.
>
> **N-Gram Definition (Q1):** To define N-grams, we follow the procedure indicated in Figure 1 of the paper: tokens are first linearized using a row-wise linearization scheme (as is done in traditional transformer approaches), giving a 1-D sequence of tokens $(x_1, x_2, …, x_n)$. N-grams are then defined analogously to natural language, with 2-grams being a sequence of all pairs of tokens (i.e. $(x_1, x_2), (x_2, x_3), (x_3, x_4)$, etc.), 3-grams being a sequence of all triplets of tokens (i.e. $(x_1, x_2, x_3), (x_2, x_3, x_4)$, etc.) and other N-grams being defined similarly.
>
> We have added a section clarifying this definition in the appendix (Appendix A.1).
>
> [1] Shengbang Tong, Ellis Brown, Penghao Wu, Sanghyun Woo, Manoj Middepogu, Sai Charitha Akula, Jihan Yang, Shusheng Yang, Adithya Iyer, Xichen Pan, et al. Cambrian-1: A fully open, vision-centric exploration of multimodal llms. arXiv preprint arXiv:2406.16860, 2024. 7

---

> ### Author Response · Authors · 2024-11-20
>
> **Broad vs. Narrow Focus (W3):** While our approach is broad-reaching in nature (and covers several angles of analysis), we also believe that such a broad foundation also has significant depth of analysis, which we hope can allow the community to build on a solid, empirically validated, statistical foundation when making modality-specific architecture choices. In this work, we begin with the premise that vision and language tokens are treated equivalently in modern architectures - so we decide to look closer at the underlying statistics of those tokens to determine if such an approach is well-motivated. Doing so, using the lens of zipf’s law, we realize that These phenomena together suggest that visual VQ-VAEs are “spreading” information between the independent tokens, rather than building compressive and compositional structures (Section 2.2). We then validate these effects by looking at token innovation rates (Section 2.3), and entropy/compressibility (Section 2.5). This leads us to ask the question: “If these tokens are not encoding at the same semantic level as ‘words’, what are they encoding?,” leading to our experiments in Section 2.6, which show that tokens encode part-level representations, and our experiments in Section 3, which show that visual sequences follow unique grammatical rules, and have unique topological structures. Together, we believe these analyses provide sufficiently deep, and fairly strong experimental motivation for unique visual architectures, an insight enabled by the linguistic analysis we perform in this paper.
>
> While we understand the value of a much narrower approach that focuses on only one of these analyses - we believe that the analyses taken together in this paper are necessary at this stage to justify further investigation and to establish a robust framework for understanding the distributional consequences of treating visual tokens like natural language tokens. Future work could build on our findings by exploring individual phenomena in depth and validating specific architectural or methodological adjustments in multimodal models.  We thank the reviewer for this suggestion and we are fully on board with such changes, and we will work to make the next version clearer in the paper so that the depth of analysis is clearer.

---

> ### Author Response · Authors · 2024-11-27
>
> We appreciate the time and effort you've invested in reviewing our work! In our rebuttal and the updated paper, we hope that we've thoroughly addressed the concerns and suggestions raised in the initial reviews. We are happy to discuss any of these points further, and we are eager to engage in more discussion during the official ICLR discussion period. If you find that all your questions have been resolved, we kindly ask you to consider reflecting this in the scores.
>
> Thank you once again for your thoughtful feedback and consideration!
>
> Best regards,
> The Authors

---

### Official Review · Reviewer_cqku · 2024-11-07

**Soundness:** 3
**Presentation:** 3
**Contribution:** 2
**Rating:** 6
**Confidence:** 4

**Summary:**

This paper looks at the statistical properties of "visual languages," where images are broken into discrete tokens like words in a sentence, used in multimodal models such as transformers. The authors explore whether these visual tokens behave similarly to natural language in terms of frequency distributions, grammatical structures, and alignment. Their key findings are that visual tokens follow a Zipf-like distribution with higher entropy, lack cohesive grammar, and mainly represent parts of objects, aligning only partially with natural language. These results suggest that visual languages have unique characteristics, which might benefit from specialized model designs to handle them effectively.

**Strengths:**

1. This paper is well-oragnized and offers a fresh perspective by treating visual tokens as discrete elements analogous to words in natural language.
2. The experiments are well-executed, with thorough empirical analysis across several datasets and tokenization methods.
3. The work has significant implications for multimodal model design, suggesting that unique features of visual tokens may require new model designs for better performance in vision-language tasks.

**Weaknesses:**

1. While the paper evaluates various tokenization methods (e.g., VQ-VAE, Chameleon), it could benefit from exploring alternative tokenization strategies, especially non-discrete or hybrid methods.
2. The study primarily relies on commonly used datasets (e.g., MS-COCO, ImageNet) that may not fully capture the diversity and complexity of visual scenes in real-world multimodal applications. Including more varied datasets with richer visual and contextual details.

**Questions:**

The authors can consider to include more varied datasets with richer visual and contextual details.

---

> ### Author Response · Authors · 2024-11-20
>
> We greatly appreciate the reviewer for taking the time to review our manuscript and provide thoughtful comments, and we appreciate that the reviewer highlights the fresh perspective, thorough empirical analysis, and significant design implications highlighted in our paper.  We would like to take this opportunity to address some of the reviewer’s questions and concerns:
>
> **Alternative Tokenization Strategies:** We strongly agree that it would be valuable to explore additional tokenization strategies, particularly hybrid or continuous tokenizers (such as LLaVA). That being said, such an extension is highly non-trivial: because the natural language techniques that we explore in this paper were developed for discrete tokens (such as are common in natural language), each would have to be extended to the continuous domain.
>
> For some of these analyses, such as entropy, continuous domain generalizations exist (such as differential entropy), however are challenging to quantify in higher dimensional spaces, and to our knowledge, it would be foundational work in probability theory to demonstrate that such entropy values are comparable to those in the discrete domain.  For other analyses, such as Benford’s law, no such continuous domain generalization exists, and would have to be derived from first-principles. While it appears that there is some intuition as to the underlying foundational principles behind Benford’s law [1], we believe that simply deriving (and demonstrating) such a continuous generalization would be a significant contribution in several communities. Similar techniques would have to be derived for other methods such Yule-Simon laws or C-PCFGs, each of which we believe would be significant individual contributions.
>
> It is possible to perform analyses on quantized spaces of the continuous domain, treating the quantized states as continuous variables, however doing so introduces significant quantization bias that can impact the outcomes. For example when analyzing entropy in quantized spaces, the resolution of quantization directly impacts the calculated entropy. Coarse quantization tends to underestimate the entropy by failing to capture the full variability of the continuous domain, while fine quantization can overfit noise in the data. Similarly, for Yule-Simon distributions, the observed frequencies of quantized states would have the potential to reflect artifacts of binning rather than true reflections of the underlying continuous distribution. Thus, the resulting power-law exponent might be systematically distorted, either attenuated or exaggerated, based on the quantization scheme used.
>
> Given these considerations, we decided to omit continuous valued tokenizers from this work, as there remain a significant number of unanswered questions that would have to be addressed (either at a theoretical level, or at a quantization-validation level), however we strongly believe that future work can make strides towards evaluating such rules in the continuous domain as well. We have further clarified this in Appendix C, to provide more context for this decision.
>
> **Datasets (W2, Q1):** While some of the datasets used in this paper may represent commonly used datasets (such as MS-COCO or ILSVRC which have a restricted object set), we further include the CC12M dataset, a dataset of 12M captioned images from the web which, we believe, covers a fairly large range of diverse images from across the web. In practice, we chose these datasets carefully to range from thoroughly in-domain for the tokenizers (ILSVRC) to diverse (CC12M) to likely out of domain (XM3600), in an attempt to capture a wide diversity of situations and tokenization applications. We are happy to consider the inclusion of additional datasets in the final version of the paper, and would appreciate recommendations for example captioned datasets that would include richer visual and contextual details.
>
> [1] Becker, Thealexa, et al. "Benford’s law and continuous dependent random variables." Annals of Physics 388 (2018): 350-381.

---

> ### Author Response · Authors · 2024-11-27
>
> We appreciate the time and effort you've invested in reviewing our work! In our rebuttal and the updated paper, we hope that we've thoroughly addressed the concerns and suggestions raised in the initial reviews. We are happy to discuss any of these points further, and we are eager to engage in more discussion during the official ICLR discussion period. If you find that all your questions have been resolved, we kindly ask you to consider reflecting this in the scores.
>
> Thank you once again for your thoughtful feedback and consideration!
>
> Best regards,
> The Authors

---

### Author Response · Authors · 2024-11-22

Dear Reviewers,

We're following up on the rebuttal for our paper, "Analyzing the Language of Visual Tokens." We appreciate the time and effort you've invested in reviewing our work.

In our rebuttal and the updated paper, we hope that we've thoroughly addressed the concerns and suggestions raised in the initial reviews. We are happy to discuss any of these points further, and we are eager to engage in more discussion during the official ICLR discussion period. If you find that all your questions have been resolved, we kindly ask you to consider reflecting this in the scores.

Thank you once again for your thoughtful feedback and consideration!

Best regards,
The Authors

---

### Meta-Review · Area_Chair_VNdE · 2024-12-20

**Metareview:**

This paper tackles an interesting problem: that of analyzing the "structure" within sequential vision models, from a linguistic point of view. The main finding of the statistical evaluation is that visual tokens follow a Zipf distribution with higher entropy and lack cohesive grammar. While the study is interesting, the work only speculates on the implications of this finding, and the finding by itself is very un-surprising. The paper received very mixed reviews, but the highest-rating reviewer shared concerns regarding the practical implications of this work. Given that I also share this major concer, I recommend this paper for rejection.

**Additional Comments On Reviewer Discussion:**

The authors answered to some of the concerns, but the main one, lying at the core of this work remains unanswered : there is no clear path about what to do with this findings. The highest-rating reviewer did not accept to champion the paper, as they also shared this concern.

---

### Decision · Program_Chairs · 2025-01-22

Reject